# Minimax optimality of convolutional neural networks for infinite dimensional input-output problems and separation from kernel methods

**Yuto Nishimura**
University of Tokyo
`yutonishimurav2@g.ecc.u-tokyo.ac.jp`

**Taiji Suzuki**
University of Tokyo
RIKEN AIP
`taiji@mist.i.u-tokyo.ac.jp`

## ABSTRACT

Recent deep learning applications, exemplified by text-to-image tasks, often involve high-dimensional inputs and outputs. While several studies have investigated the function estimation capabilities of deep learning, research on dilated convolutional neural networks (CNNs) has mainly focused on cases where input dimensions are infinite but output dimensions are one-dimensional, similar to many other studies. However, many practical deep learning tasks involve high-dimensional (or even infinite dimensional) inputs and outputs. In this paper, we investigate the optimality of dilated CNNs for estimating a map between infinite-dimensional input and output spaces by analyzing their approximation and estimation abilities. For that purpose, we first show that approximation and estimation errors depend only on the smoothness and decay rate with respect to the infinity norm of the output, and their estimation accuracy actually achieve the *minimax optimal* rate of convergence. Second, we demonstrate that the dilated CNNs outperform *any* linear estimators including kernel ridge regression and $k$-NN estimators in a minimax error sense, highlighting the usefulness of feature learning realized by deep neural networks. Our theoretical analysis provide a theoretical basis for understanding the success of deep learning in recent high-dimensional input-output tasks.

## 1 INTRODUCTION

In recent years, deep learning has found applications in a wide array of fields, leading to remarkable progress. Some noteworthy breakthroughs include Stable Diffusion (Rombach et al. (2022)) for text-to-image applications, and Whisper (Radford et al. (2022)) for speech-to-text applications. Despite their distinctiveness, these technologies have one thing in common: They handle high-dimensional inputs and outputs. This is quite different from the early days of deep learning, which began with AlexNet (Krizhevsky et al. (2017)) and mainly focused on classification tasks with high-dimensional inputs but only one-dimensional outputs.

The theoretical research of deep learning has studied the approximation and estimation capabilities of neural networks. It is well-established that two-layer neural networks can approximate any continuous function with compact support to arbitrary precision (Cybenko (1989); Hornik (1991)), while multi-layer networks have been investigated under more practical conditions, such as functions belonging to Hölder or Besov spaces (Petersen & Voigtlaender (2018); Suzuki (2019)). Estimation capabilities of multi-layer networks have also been examined using a finite number of samples, with some studies showing nearly minimax optimal convergence rates (Schmidt-Hieber (2020)).

However, these studies often assume a fixed input dimension significantly smaller than the size of training data, resulting in the curse of dimensionality where convergence rates depend on the input data dimension. This limitation is particularly relevant for recent deep learning tasks, such as processing very long textual, image and audio data, where input dimensions can be extremely large or even regarded as infinite-dimensional.

To overcome this challenge, researchers have focused on settings where the data distribution's support exhibits low-dimensional structures (Chen et al. (2019; 2022); Nakada & Imaizumi (2020)). These studies demonstrate that neural networks can avoid the curse of dimensionality by exploiting low-dimensional data structures. Suzuki & Nitanda (2021) demonstrated the ability to overcome the curse of dimensionality even when the input data does not lie on a low-dimensional manifold by considering Besov spaces with direction-dependent smoothness, referred to as anisotropic Besov spaces. Nevertheless, the performance of neural networks in extremely high-dimensional or infinite-dimensional settings is still an open question that warrants further investigation.

Okumoto & Suzuki (2021) considered $\gamma$-*smooth space* (Def. 1), demonstrating that dilated CNNs can achieve approximation and estimation errors that depend only on smoothness, not dimensionality, for infinite-dimensional inputs. The $\gamma$-smooth space is a function space where different coordinates exhibit different smoothness, which is inspired by the settings in Dũng & Griebel (2016); Ingster & Stepanova (2011). This research partially explains the great success of dilated CNNs in tasks involving high-dimensional inputs. However, including this study, theoretical research on deep learning has primarily focused on one-dimensional outputs, neglecting the situation with high-dimensional outputs. Given that many recent deep learning applications involve high-dimensional inputs and outputs, it is essential to conduct theoretical analyses of neural networks in problem settings where both input and output dimensions are high.

In recent research, while there has been limited investigation on deep learning in the context of infinite-dimensional inputs and outputs, the study of linear operators in such settings has been extensively explored. For example, Oliva et al. (2013; 2014) proposed a method for estimating mappings with inputs and outputs as functions or distributions, and showed their convergence rates. In Talwai et al. (2022), the authors provided convergence rates for estimating linear operators with input and output spaces being reproducing kernel Hilbert spaces, extending the results of Fischer & Steinwart (2020) for one-dimensional outputs. Jin et al. (2022) demonstrated a more general form of linear operator estimation errors by changing the norm used in Talwai et al. (2022), reflecting the structure of output data spaces. Research on learning a linear operator in infinite-dimensional input-output settings has also been widely studied in the literature of numerical analysis (Lu et al. (2021); Li et al. (2021b); de Hoop et al. (2021); Li et al. (2018; 2021a)) and econometrics (Singh et al. (2019); Muandet et al. (2020); Dikkala et al. (2020)).

Additionally, there are several studies that have conducted analysis using neural networks targeting nonlinear operators with infinite-dimensional input and output spaces. Chen & Chen (1995) proved that neural networks have the capability of approximating nonlinear functionals defined on some compact set of a Banach space and nonlinear operators. Then, Lu et al. (2021) extended the theory to deep neural networks and proposed DeepONet based on the theory. Lanthaler et al. (2022) conducted a theoretical analysis of DeepONet, providing its approximation and estimation errors. However, their analysis is contingent upon the architecture of DeepONet and is confined to a limited scope, specifically focusing on learning operators between functions.

**Our contributions.** In this study, we show the optimality of dilated CNNs by analyzing their approximation and estimation errors in a problem setting where the input and output dimensions are both *infinite*, while existing work did not show the statistical optimality of this problem. We deal with the problem as a nonparametric regression problem with infinite dimensional input-output spaces where the true function is a nonlinear operator consisting of an infinite sequence of functions belonging to the so-called $\gamma$-*smooth space* (Dũng & Griebel, 2016; Ingster & Stepanova, 2011; Okumoto & Suzuki, 2021). The intuition behind this function class can be explained via practical applications such as audio-data conversion. When we convert audio-data, the input and output audio data are usually decomposed into the frequency domain (or other audio features such as Mel-frequency cepstrum) and the input-output relation is often highly sensitive to some frequency bands, which can be formalized by the non-uniform smoothness with respect to each frequency component. The $\gamma$-smooth space provides a mathematical abstraction of this notion. Then, our purpose is to explicitly verify how advantageous the CNN structure is to estimate this kind of infinite dimensional non-linear dependency. More specifically, our contributions can be summarized as follows:

1. We consider a nonlinear operator as a true function with infinite-dimensional inputs and outputs. In the aforementioned setup, we demonstrate that dilated CNNs achieve approximation and estimation errors that depend only on the smoothness and decay rate of the output. Furthermore,

we show that the estimation errors are minimax optimal. To the best of our knowledge, this is the first study that establishes the minimax optimality of deep learning models in the setting of infinite-dimensional input and output. Technically, we show lower bounds on the minimax optimal rate by providing a covering number of spaces that have decay in its $L^2$ norms of the outputs, and this framework can be applicable to other function classes as well.

2. We show that dilated CNNs are adaptive to the unknown smoothness structure, that is, it automatically achieves the minimax rate without the knowledge of the smoothness structure of the true target functional. To show how crucial this adaptivity is in terms of the predictive error, we compare the predictive performance with the class of *linear estimators* as a counter part of non-feature learning methods including kernel ridge regression and $k$-NN estimator. Indeed, we show that the worst case error of *any* linear estimator is outperformed by dilated CNNs. This result highlights the usefulness of neural network-based feature extraction.

These results demonstrate that dilated CNNs achieve the minimax optimality with polynomial order sample complexity, even when both the input and output dimensions are infinite-dimensional. Furthermore, dilated CNNs are found to be superior to linear estimators. These findings underscore the significance of dilated CNNs' feature extraction abilities. This theoretical analysis partially explains the success of deep learning in recent high-dimensional input-output tasks.

## 2 PROBLEM SETTING AND NOTATIONS

In this section, we set up our problem setting of the nonlinear operator learning. First, we explain the notation used throughout this paper. Let $\mathbb{R}_{>0} := \{s \in \mathbb{R} : s > 0\}$, and for a set $\mathbb{D}$, let $\mathbb{D}^\infty := \{(s_1, \ldots, s_i, \ldots) : s_i \in \mathbb{D}\}$. For $s \in \mathbb{D}^\infty$, we define $\mathrm{supp}(s) := \#\{i \in \mathbb{N} : s_i \neq 0\}$. We then define $\mathbb{N}_0^\infty := \{l \in (\mathbb{N} \cup \{0\})^\infty : \mathrm{supp}(l) < \infty\}$ and similarly define $\mathbb{R}_0^\infty$, $\mathbb{Z}_0^\infty$. Furthermore, for $s \in \mathbb{N}_0^\infty$, we let $2^s := 2^{\sum_{i=1}^\infty s_i}$. For $a \in \mathbb{R}$, $\lfloor a \rfloor$ is the largest integer less than or equal to $a$, and $\lceil a \rceil$ is the smallest integer greater than or equal to $a$. Additionally, for $L \in \mathbb{N}$, we let $[L] := \{1, \ldots, L\}$.

**Problem setting** Let $\lambda$ be the Lebesgue measure on $([0,1], \mathcal{B}([0,1]))$, where $\mathcal{B}([0,1])$ is the Borel $\sigma$-field on $[0,1]$. Let $\lambda^\infty$ be the product measure defined on $([0,1]^\infty, \prod_{i=1}^\infty \mathcal{B}([0,1]))$ obtained as the countably infinite product of $\lambda$. Such measures exist uniquely by the Kolmogorov extension theorem. Let $\mathcal{M} := [0,1]^\infty$. Let $P_X$ be a probability measure defined on $\mathcal{M}$ that is absolutely continuous with respect to $\lambda^\infty$, and let its Radon-Nikodym derivative satisfy $\left\| \frac{\mathrm{d}P_x}{\mathrm{d}\lambda^\infty} \right\|_{L^\infty(\mathcal{M})} < \infty$. Then, we assume that there exists a true nonlinear operator $f^\circ : \mathcal{M} \to \mathbb{R}^\infty$ such that the input and output have the following nonlinear relation: (1) $Y = f^\circ(X) + \xi$, where $X$ is a random variable that takes values in $\mathcal{M}$ obeying the distribution $P_X$, and $\xi \in \mathbb{R}^\infty$ is an observation noise such that each component is independently following a Gaussian distribution with mean 0 and bounded variance. In this study, we discuss (i) how efficiently neural networks can approximate the true operator $f^\circ$, and (ii) how accurately neural networks can estimate the true operator $f^\circ$ from $n$ observation data $D_n = \left(x^{(i)}, y^{(i)}\right)_{i=1}^n \subset \mathcal{M} \times \mathbb{R}^\infty$. We use the mean squared error $\|f - f^\circ\|_{P_X}^2 := \mathbb{E}_P\left[\|f(x) - f^\circ(x)\|_{\ell^2}^2\right] = \mathbb{E}_P\left[\sum_{j=1}^\infty \left(f_j(x) - f_j^\circ(x)\right)^2\right]$ as a performance metric for learning operators. Here, $P$ denotes the joint distribution of the random variables $X$ and $Y$.

**Notations.** Next, we define the $\gamma$-smooth space (Okumoto & Suzuki (2021)). For $l \in \mathbb{Z}_0^\infty$, $x = (x_i)_{i=1}^\infty \in \mathcal{M}$, we define

$$\psi_l(x) := \prod_{i=1}^\infty \psi_{l_i}(x_i), \quad \psi_{l_i}(x_i) := \begin{cases} \sqrt{2}\cos(2\pi|l_i|x_i) & (l_i < 0), \\ \sqrt{2}\sin(2\pi|l_i|x_i) & (l_i > 0), \\ 1 & (l_i = 0). \end{cases}$$

Then, let $L^2(\mathcal{M}) := \left\{f : \mathcal{M} \to \mathbb{R} : \int_\mathcal{M} f^2(x)\,\mathrm{d}\lambda^\infty(x) < \infty\right\}$ equipped with an inner product $\langle f, g \rangle := \int_\mathcal{M} f(x)g(x)\,\mathrm{d}\lambda^\infty(x)$ for $f, g \in L^2(\mathcal{M})$. Here, the set $(\psi_l)_{l \in \mathbb{Z}_0^\infty}$ forms an orthonormal basis of this space (Ingster & Stepanova, 2011), and thus any $f \in L^2(\mathcal{M})$ can be expanded as $f(\cdot) = \sum_{l \in \mathbb{Z}_0^\infty} \langle f, \psi_l \rangle \psi_l(\cdot)$. For $s \in \mathbb{N}_0^\infty$, we define $\delta_s(f) : \mathbb{R}_0^\infty \to \mathbb{R}$ by $\delta_s(f)(\cdot) := \sum_{l \in \mathbb{Z}_0^\infty : \lfloor 2^{s_i-1} \rfloor \leq |l_i| < 2^{s_i}} \langle f, \psi_l \rangle \psi_l(\cdot)$, which can be interpreted as the frequency components of $f$ corresponding to frequency of $|l_i| \sim 2^{s_i}$. Let $J(s) := \left\{l \in \mathbb{Z}_0^\infty : \lfloor 2^{s_i-1} \rfloor \leq |l_i| < 2^{s_i}\right\}$.

For $p \geq 1$, if we define $\|f\|_p := \left( \int_{\mathcal{M}} |f|^p \, \mathrm{d}\lambda^{\infty} \right)^{\frac{1}{p}}$, then $\gamma$-smooth space can be defined as follows:

**Definition 1** ($\gamma$-smooth space). Let $p, q \geq 1$, and $\gamma : \mathbb{N}_0^{\infty} \to \mathbb{R}$ be a monotonically increasing function for each component. Then, the $\gamma$-smooth space is defined as $\mathcal{F}_{p,q}^{\gamma} := \left\{ f \in L^2(\mathcal{M}) : \|f\|_{\mathcal{F}_{p,q}^{\gamma}} < \infty \right\}$, whose norm is $\|f\|_{\mathcal{F}_{p,q}^{\gamma}} := \left( \sum_{s \in \mathbb{N}_0^{\infty}} \left( 2^{\gamma(s)} \|\delta_s(f)\|_p \right)^q \right)^{\frac{1}{q}}$.

We can see that the function $\gamma$ plays a kind of penalty on each frequency component of the functions in the class; that is, a frequency component with large $\gamma(s)$ should be suppressed so that the norm is bounded. In that sense, the design of $\gamma(s)$ is crucial to control their smoothness. In this study, we focus on two types of $\gamma$ functions as in Okumoto & Suzuki (2021).

**Definition 2** (Mixed smoothness and anisotropic smoothness). Given a sequence $a = (a_i)_{i=1}^{\infty} \in \mathbb{R}_{>0}^{\infty}$, *mixed smoothness* is defined as $\gamma(s) = \langle a, s \rangle$, where $\langle a, s \rangle = \sum_{i=1}^{\infty} a_i s_i$. Moreover, *anisotropic smoothness* is defined as $\gamma(s) = \max_{i \in \mathbb{N}} \{a_i s_i\}$.

Intuitively, $a$ represents the strength of penalty on each frequency. Indeed, each $a_i$ can be interpreted as the smoothness toward the $i$-th coordinate. More technically, it is known that when $\gamma$ has mixed smoothness or anisotropic smoothness, the $\gamma$-smooth space becomes an extension of mixed Besov space (Schmeisser (1987)) or anisotropic Besov space (Nikol'skii (1975)) to an infinite dimensional settings, respectively (Okumoto & Suzuki (2021)).

Based on the $\gamma$-smooth function class, we define a class of nonlinear operators from $\mathcal{M}$ to $\mathbb{R}^{\infty}$. First, let $\left( L^2(\mathcal{M}) \right)^{\infty} := \left\{ f : \mathcal{M} \to \mathbb{R}^{\infty} : \int_{\mathcal{M}} \|f(x)\|_{\ell^2}^2 \, \mathrm{d}\lambda^{\infty}(x) < \infty \right\}$ equipped with an inner product $\langle f, g \rangle := \int_{\mathcal{M}} \sum_{i=1}^{\infty} (f(x))_i (g(x))_i \, \mathrm{d}\lambda^{\infty}(x)$ by an abuse of notation. Then we define a class of operators $\left( \mathcal{F}_{p,q}^{\gamma} \right)^{\infty}$ as follows: $\left( \mathcal{F}_{p,q}^{\gamma} \right)^{\infty} := \left\{ f \in \left( L^2(\mathcal{M}) \right)^{\infty} : \forall i \in \mathbb{N}, \ \|f_i\|_{\mathcal{F}_{p,q}^{\gamma}} < \infty \right\}$, which is an extension of the $\gamma$-smooth space to an infinite dimensional output setting. In this study, we mainly discuss nonlinear operators that belong to this space.

**Assumptions.** Next, we impose assumptions on the true nonlinear operator for theoretical analysis in this study. The following assumption imposes a norm control on each component $f_i^{\circ}$ of the target function:

**Assumption 3.** We assume that the true nonlinear operator $f^{\circ}$ satisfies the following condition for some $p \geq 1$, $q \geq 1$[1]:

$$f^{\circ} = (f_i^{\circ})_{i=1}^{\infty} \in \left( U\left( \mathcal{F}_{p,q}^{\gamma} \right) \right)^{\infty}, \tag{2}$$

where $\gamma$ is the mixed or anisotropic smoothness. Furthermore, there exist constants $B_2 > 0$, $B_{\infty} > 0$ and $0 < r < 1$ such that

$$\|f_i^{\circ}\|_2 \leq B_2 i^{-\frac{1}{r}}, \ \|f_i^{\circ}\|_{\infty} \leq B_{\infty} \quad \left( \forall i \in \mathbb{N} \right). \tag{3}$$

We let $\mathcal{B}_r$ be the subset of $(L^2(\mathcal{M}))^{\infty}$ satisfying the condition (3).

Here, we define the $L^p$-norm as $\|f\|_p := \left( \int_{\mathcal{M}} \|f(x)\|_{\ell^p}^p \, \mathrm{d}\lambda^{\infty}(x) \right)^{\frac{1}{p}}$ for $f : \mathcal{M} \to \mathbb{R}^{\infty}$ by an abuse of notation. In particular, we set $\|f\|_{\infty} := \sum_{i=1}^{\infty} \|f_i\|_{\infty}$ when $p = \infty$. Note that $\|\cdot\|_2 \leq \|\cdot\|_{\infty}$. Since $P_X$ satisfies the assumption of absolute continuity and the finiteness of the Radon-Nikodym derivative with respect to $\lambda^{\infty}$, we have $\|\cdot\|_{P_X} \lesssim \|\cdot\|_2$ (Okumoto & Suzuki (2021)). We also impose the following assumption on the observation noise:

**Assumption 4.** Each component of the observation noise $\xi = (\xi_i)_{i=1}^{\infty} \in \mathbb{R}^{\infty}$ is assumed to be independent and follow a Gaussian distribution as $\xi_i \sim \mathcal{N}\left( 0, \sigma_i^2 \right)$ where $\sigma_i^2$ is uniformly bounded by $\bar{\sigma}^2 < \infty$.

These assumptions posit that the true function values tend to approach zero as the index increases. Without such assumptions, there would be issues such as the accumulation of estimation errors in the outputs corresponding to each index leading to divergence. It is still a future task to weaken the assumptions by changing the norm used for evaluation, as in Jin et al. (2022). An intuition behind this assumption can be obtained by considering a speech synthesis task, for example. Suppose that the output data is decomposed into the frequency components. Then, the index $i$ corresponds to $i$-th frequency. The assumptions above corresponds to a situation where the amplitude of the true signal decreases as the frequency increases.

---

[1] We denote by $U(\mathcal{X})$ the unit ball of a normed vector space $\mathcal{X}$.

## 3 APPROXIMATION AND ESTIMATION ERRORS OF DILATED CNNS

In this section, we analyze the approximation and estimation errors in learning nonlinear operators using dilated convolutional neural networks (CNNs). First, we define the dilated CNNs, which is a model that consists of convolutional layers followed by fully connected layers. We begin by defining the fully connected neural network (FNNs).

**Definition 5** (Fully connected neural network (FNNs)). Let $L \in \mathbb{N}$ be the depth of the network, and $i = 1, 2, \ldots, L+1$, $d_i \in \mathbb{N}$ be the width of the $i$-th layer. Then, the model defined by $f(x) = (A_L \eta(\cdot) + b_L) \circ \cdots \circ (A_i \eta(\cdot) + b_i) \circ \cdots \circ (A_1 x + b_1)$, where $\eta(\cdot)$ is the ReLU activation function and $A_i \in \mathbb{R}^{d_{i+1} \times d_i}$, $b_i \in \mathbb{R}^{d_{i+1}}$, $\eta(x) = \max\{x, 0\}$. This model is called the **fully connected neural network (FNNs)**.

We also consider the following collection of FNNs using some constants $W \in \mathbb{N}$, $S \in \mathbb{N}$, $B > 0$:

$$\Phi(L, W, S, B) := \Big\{ f(x) = (A_L \eta(\cdot) + b_L) \circ \cdots \circ (A_i \eta(\cdot) + b_i) \circ \cdots \circ (A_1 x + b_1) :$$

$$\max_{i=1,\ldots,L} \|A_i\|_\infty \vee \|b_i\|_\infty \leq B, \ \sum_{i=1}^{L} \|A_i\|_0 + \|b_i\|_0 \leq S, \ \max_{i=1,\ldots,L+1} d_i \leq W \Big\},$$

where $\|\cdot\|_\infty$ returns the maximum absolute value of a vector/matrix and $\|\cdot\|_0$ returns the number of nonzero elements of a vector/matrix.

This collection of FNNs is an extension of the collection of FNNs for one-dimensional output analyzed in Bölcskei et al. (2019), Suzuki (2019), and Schmidt-Hieber (2020) to the case of multi-dimensional output. Therefore, unlike these previous studies, the assumption that $d_{L+1} = 1$ is not made, and it is treated similarly to other variables.

Next, we define the dilated CNNs. Let $C \in \mathbb{N}$ be the number of channels. We define $\mathbb{R}^{C \times \infty} := \big\{ (x_1, \ldots, x_i, \ldots) : x_i \in \mathbb{R}^C \big\}$. Let $W' \in \mathbb{N}$ be the filter width, and $w \in \mathbb{R}^{C \times W'}$ be a filter and $X = (x_{i,j})_{i=1,j=1}^{C,\infty} \in \mathbb{R}^{C \times \infty}$ be the input. Then, for an interval $h \in \mathbb{N}$, we define the dilated convolution $w \star_h X \in \mathbb{R}^\infty$ as $(w \star_h X)_k := \sum_{i=1}^{C} \sum_{j=1}^{W'} w_{i,j} x_{i,h(j-1)+k}$. Note that this coincides with the usual convolution when $h = 1$. Here, let $C' \in \mathbb{N}$ be the number of output channels, and $F \in \mathbb{R}^{C' \times C \times W'}$ be a filter. Then, we define $\mathrm{Conv}_{h,F} : \mathbb{R}^{C \times \infty} \to \mathbb{R}^{C' \times \infty}$ as $\mathrm{Conv}_{h,F}(X) := (F_{1,:,:} \star_h X, \cdots, F_{C',:,:} \star_h X)^T$. Using these notations, we define the dilated CNNs as follows.

**Definition 6** (dilated CNNs). Given constants $L', W' \in \mathbb{N}$, and filters $F_l \in \mathbb{R}^{C_{l+1} \times C_l \times W'}$ with channel numbers $C_l \in \mathbb{N}$ for $l \in [L']$, where $C_1 = 1$. Then, we define the **dilated CNNs** as $f(X) = g_{\mathrm{FNN}} \circ \Big( \mathrm{Conv}_{W'^{L'-1}, F_{L'}} \circ \cdots \circ \mathrm{Conv}_{W'^{l-1}, F_l} \circ \cdots \circ \mathrm{Conv}_{1, F_1} \circ X \Big)_1$, where $g_{\mathrm{FNN}} \in \Phi(L, W, S, B)$ is a FNNs with input $x \in \mathbb{R}^{C_{L'}}$ and output $g_{\mathrm{FNN}}(x) \in \mathbb{R}^{d_{L+1}}$. In other words, it is important to note that $f$ is a function from the input space $\mathcal{M}$ to $\mathbb{R}^{d_{L+1}}$.

We also define the set of dilated CNNs with a constant $B' > 0$, $C \in \mathbb{N}$ as follows:

$$\mathcal{P}(L', B', W', C, L, W, S, B) := \Big\{ g_{\mathrm{FNN}} \circ \Big( \mathrm{Conv}_{W'^{L'-1}, F_{L'}} \circ \cdots \circ \mathrm{Conv}_{1, F_1} \circ X \Big)_1 :$$

$$F_l \in \mathbb{R}^{C \times C \times W'} \ (l \geq 2), \ F_1 \in \mathbb{R}^{C \times 1 \times W'}, \ \|F_l\|_\infty \leq B', \ g_{\mathrm{FNN}} \in \Phi(L, W, B, S) \Big\},$$

where we assumed $C_l = C$ $(l \geq 2)$ to fix the number of channels in each layer.

When considering estimation errors, it becomes important to make use of boundedness of the estimated function in terms of $\|\cdot\|_\infty$. Therefore, we define the set of *bounded CNNs* with $B_2 > 0$ as $\bar{\mathcal{P}} := \bar{\mathcal{P}}(B_2, L', B', W', C, L, W, S, B)$, where

$$\bar{\mathcal{P}} = \left\{ (\bar{f}(X))_i = \begin{cases} -B_2 i^{-\frac{1}{r}} \vee \left( B_2 i^{-\frac{1}{r}} \wedge (f(X))_i \right) & (1 \leq i \leq d_{L+1}) \\ 0 & (d_{L+1} < i) \end{cases} : f \in \mathcal{P} \right\},$$

where $\mathcal{P} = \mathcal{P}(L', B', W', C, L, W, S, B)$. Here, $f \in \bar{\mathcal{P}}$ is a function $f : \mathcal{M} \to \mathbb{R}^\infty$, which naturally extends the output of $f \in \mathcal{P}$ from $d_{L+1}$ to $\infty$.

### 3.1 APPROXIMATION ERROR OF DILATED CNNS

Here, we analyze the approximation error of nonlinear operators using dilated CNNs. In the case of the learning of functions with one-dimensional output ($f^\circ \in \mathcal{F}_{p,q}^\gamma$), the approximation error is shown in Okumoto & Suzuki (2021). Here, we extend their results to infinite dimensions.

Before doing so, we introduce the following definition for $a$.

**Definition 7** (Definition of smoothness). (i) A sequence $\boldsymbol{a}$ is said to *increase in polynomial order* if $a = (a_i)_{i=1}^\infty$ is a monotonically increasing sequence of positive real numbers, and there exists a constant $\eta > 0$ such that $a_i = \Omega(i^\eta)$, and $a_1 < a_2$. (ii) Furthermore, $\boldsymbol{a}$ is said to have *sparsity* if $a = (a_i)_{i=1}^\infty$ is a sequence of positive real numbers satisfying $a_i = \Omega(\log i)$, and there exists a constant $\eta > 0$ such that the sequence $(a_{i_j})_{j=1}^\infty$ obtained by sorting $\{a_i\}_{i=1}^\infty$ in ascending order satisfies $\|a\|_{wl^\eta} := \sup_j j^\eta a_{i_j}^{-1} \leq 1$.

The first setting is the easiest setting where the "importance" of each coordinate monotonically increases as the index increase, where the importance is measured by the smoothness $a_i$. On the other hand, the second setting is more challenge where we need to find important features with small $a_i$ from wide range of coordinates. The theorem including this assumption for the approximation error is shown below.

**Theorem 1** (Approximation error of nonlinear operators by dilated CNNs). Suppose the true nonlinear operator $f^\circ \in \left(\mathcal{F}_{p,q}^\gamma\right)^\infty$ satisfies Assumption 3. If $a$ increase in polynomial order, we set $L' = 1$, $B' = 1$, $W' \sim T^{\frac{1}{\eta}}$, $C \sim T^{\frac{1}{\eta}}$ and if $a$ has sparsity, we set $L' \sim T$, $B' = 1$, $W' = 3$, $C \sim T^{\frac{1}{\eta}}$. (i) (mixed smoothness) If $\gamma$ has mixed smoothness, let $a^\dagger = a_{i'}$, where $i' = 1$ when $a$ increases in polynomial order, and $i' = i_1$ when $a$ has sparsity; (ii) (anisotropic smoothness) If $\gamma$ has anisotropic smoothness, let $a^\dagger = \left(\sum_{i=1} a_i^{-1}\right)^{-1}$. Let $v := \max\left\{\frac{1}{p} - \frac{1}{2}, 0\right\}$, and assume $v < a^\dagger$. Then, for any $T > 0$, we set $L(T) \sim \max\left\{T^{\frac{2}{\eta}}, T^2\right\}$, $W(T) \sim \max\left\{T^{\frac{1}{\eta}}2^{\frac{T}{a^\dagger}}, 2^{r(1 - \frac{v}{a^\dagger})T}\right\}$, $S(T) \sim T^{\frac{2}{\eta}}2^{r(1 - \frac{v}{a^\dagger})T + \frac{T}{a^\dagger}}$, $B(T) \sim 2^{(T/2)^{\frac{1}{\eta}}}$. Then, there exists a dilated CNNs $f' \in \bar{\mathcal{P}}(B_2, L', B', W', C, L(T), W(T), S(T), B(T))$ such that the following evaluation is obtained:
$$\|f' - f^\circ\|_2 \lesssim 2^{-(1 - r/2)(1 - v/a^\dagger)T}.$$

Unlike existing results such as Okumoto & Suzuki (2021), this theorem covers a setting with infinite dimensional output. As a result, there appears following difference: (i) The first difference is the rate of convergence. For example, the result in Okumoto & Suzuki (2021) is $\|f' - f^\circ\|_2 \lesssim 2^{-(1 - v/a^\dagger)T}$, while in this study it is $\|f' - f^\circ\|_2 \lesssim 2^{-(1 - r/2)(1 - v/a^\dagger)T}$, which increases by a factor of $r$. (ii) The second difference is the setting of the parameters $W$ and $S$ in the neural network, where the number of weights in the output layer, $2^{r(1 - v/a^\dagger)T}$, is newly included compared to Okumoto & Suzuki (2021). In both cases, these differences arise from the requirement to increase the output dimension $d_{L+1}$ significantly, such that $\sum_{i=d_{L+1}+1}^\infty \|f_i\|_2^2$ can be negligible in comparison to the other errors. Despite these differences, the core message from the theory remains the same: by assuming the decay of the output, the approximation error can be of polynomial order even when the output is in an infinite-dimensional space.

We also discuss the differences between existing work on estimating linear operators and our approach. According to Fischer & Steinwart (2020), the rate of approximation error for a function in a certain RKHS with respect to the true function $f_P^* := \mathbb{E}[Y|X = x]$ defined on the data space $X \times Y$ is given as $\lambda^{\beta - \gamma}$ (Lemma 14 of Fischer & Steinwart (2020)). On the other hand, Talwai et al. (2022) extended this result to the case where the true function is represented by a linear operator $C_{Y|X} : \mathcal{H}_K \to \mathcal{H}_L$. In this case, the approximation error is given as $\lambda^{\frac{\beta - \gamma}{2}}$ (Lemma 6 of Talwai et al. (2022)). Here, it is assumed that $\lambda$ is sufficiently smaller than 1 and $\beta - \gamma > 0$. Therefore, it can be said that the rate deterioration due to the extension to infinite dimensions occurs in the case of linear operators, similar to our results. Although the specific changes may differ due to varying assumptions and problem settings, the shared characteristic of deterioration in both cases is an intriguing implication.

## 3.2 ESTIMATION ERROR OF DILATED CNNS

Here, we analyze the estimation error of dilated CNNs based on the approximation error analysis in the last section. Now, suppose that we are given $n$ observation data points $D_n = \left(x^{(i)}, y^{(i)}\right)_{i=1}^n$ following the model (1). We consider the following empirical risk minimization (ERM) estimator: $\hat{f} \in \operatorname{argmin}_{f \in \bar{\mathcal{P}}} \frac{1}{n} \sum_{i=1}^n \|f(x_i) - y_i\|_{\ell^2}^2$. As mentioned earlier, we use the mean squared error $\left\|\hat{f} - f^\circ\right\|_{P_X}^2$ as the evaluation metric for this estimator. Since $\hat{f}$ depends on the data $D_n$, we calculate the expected value with respect to $D_n$: $\mathbb{E}_{P^n}\left[\left\|\hat{f} - f^\circ\right\|_{P_X}^2\right] := \mathbb{E}_{\left(x^{(i)}, y^{(i)}\right)_{i=1}^n \sim P^n}\left[\left\|\hat{f} - f^\circ\right\|_{P_X}^2\right]$.

First, we give a lower bound of the minimax optimal rate in the following theorem.

**Theorem 2** (Minimax optimal rate for estimating a function in the $\left(\mathcal{F}_{p,q}^\gamma\right)^\infty$). Assume that $p \geq 2$, $P_X$ is a uniform distribution over $\mathcal{M}$. In accordance with the definition used in Theorem 1, we use the same notation. Then, the minimax optimal rate is lower bounded as follows:

$$\inf_{\hat{f}} \sup_{f^* \in (U(\mathcal{F}_{p,q}^\gamma))^\infty \cap \mathcal{B}_r} \mathbb{E}_{D_n}\left[\left\|\hat{f} - f^*\right\|_{L^2(P_X)}^2\right] \gtrsim n^{-\frac{(2-r)a^\dagger}{2a^\dagger + 1}},$$

where "inf" is taken over all estimators based on $D_n$ and the expectation is taken for the sample distribution.

We see that the minimax rate is merely characterized by the smoothness parameter $a^\dagger$ and the decay rate $r$. To prove this theorem, we utilized the classic information theoretic lower bound (Yang & Barron, 1999; Raskutti et al., 2012). To do so, we carefully evaluated the covering number of the function space by taking the decay of $L^2$-norms $(\|f_i^\circ\|_2)_{i=1}^\infty$ into account, which is very unique to our problem. We believe that this framework can be extended to Cartesian product spaces consisting of other function spaces as well.

Then, we derive the upper bound of the estimation error of the ERM estimator on the class of dilated CNNs and see its optimality.

**Theorem 3** (Estimation error of nonlinear operators by dilated CNN). Assume that the true nonlinear operator $f^\circ \in \left(\mathcal{F}_{p,q}^\gamma\right)^\infty$ satisfies Assumption 3. If $a$ increases in polynomial order, set $L' = 1$, $B' = 1$, $W' \sim (\log n)^{\frac{1}{\eta}}$, $C \sim (\log n)^{\frac{1}{\eta}}$, Otherwise, if $a$ has sparsity, set $L' \sim \log n$, $B' = 1$, $W' = 3$, $C \sim (\log n)^{\frac{1}{\eta}}$. In accordance with the definition used in Theorem 1, we use the same notation for $a^\dagger, v, L(T), W(T), S(T)$, and $B(T)$. For $T \sim \log_2 n$, let $(L, W, S, B) = (L(T), W(T), S(T), B(T))$. Then, the ERM estimator $\hat{f}$ achieves the following estimation error:

$$\mathbb{E}_{P^n}\left[\left\|\hat{f} - f^\circ\right\|_{P_X}^2\right] \lesssim n^{-\frac{(2-r)\left(a^\dagger - v\right)}{2(a^\dagger - v) + 1}} (\log n)^{\frac{2}{\eta} + 2} \max\left\{(\log n)^{\frac{4}{\eta}}, (\log n)^4\right\}.$$

Because of this theorem, we can see that dilated CNNs can achieve the minimax optimal rate in the regime of $p \geq 2$ (i.e., $v = 0$) up to poly-log order. Comparing with the single output setting (Theorem 10 in Okumoto & Suzuki (2021)), we can see that there appears $r$ in the rate of convergence while that for the single output achieves $n^{-2(a^\dagger - v)/(2(a^\dagger - v) + 1)}\operatorname{polylog}(n)$. As $r$ increases (i.e., the decay is slower), the rate becomes slower, which is intuitively natural because we need to estimate more output functions accurately leading to more difficult problem. We emphasize that our result shows the minimax optimality of dilated CNNs including the effect of $r$, which is not trivial.

Lanthaler et al. (2022) provided an upper bound on the estimation error using DeepONet, solving infinite-dimensional problems based on the assumption of Lipschitz continuity, with the additional condition of input decay. Their approach relies on the assumption of the DeepONet model. In contrast, we demonstrated estimation errors in a more general and widely applicable context using dilated CNNs. Importantly, we established that our approach shows the minimax optimality.

According to these theorems, we found that, dilated CNNs can achieve *dimension independent* estimation error depending only on $a^\dagger$. It means that dilated CNNs can extract the feature whose direction has small values of $a_i$ even when $(a_i)_{i=1}^\infty$ is not monotonically sorted. This is due to the

feature extraction ability of dilated CNNs that can adatipvely find important features by training data. In the following section, we will discuss how this intuition serves to contrast dilated CNNs with linear estimators such as kernel ridge regression.

**Limitations.** One of the limitations of our research is that all of the aforementioned theorems depend on the assumption that $p \geq 2$. This arises directly from the proof of Lemma 11 in Appendix B.2, specifically from the evaluation of $\delta_s(f)$. The difficulty in evaluating $\delta_s(f)$ has also impacted prior research. For instance, in Okumoto & Suzuki (2021), which presented the approximation error of one-dimensional output in dilated CNNs, the term $v$ used in our study is incorporated due to this evaluation. In this study, we consistently used the assumption $p \geq 2$ to align with previous research, but revisiting this assumption will be part of our future work.

## 4 COMPARISON WITH LINEAR ESTIMATORS

In this section, we compare the estimation error of the dilated CNNs obtained in the previous section with those of the linear estimators (Korostelev & Tsybakov (1993); Imaizumi & Fukumizu (2019)). An estimator $\hat{f}$ based on $(x^i, y^i)_{i=1}^n$ is said to be *linear* if it can be expressed in the form of $\left(\hat{f}(x)\right)_j = \sum_{i=1}^n y_j^{(i)} \varphi_{j,i}(x; x^n) \;\; (\forall j \in \mathbb{N})$. This estimator class includes several practical estimators such as kernel ridge regression and the Nadaraya–Watson estimator; indeed, kernel ridge estimator is given as $\hat{f}(x) = K_x (K + \lambda I)^{-1} Y$ which is linear to $Y$. Then, we demonstrate that under certain conditions, the dilated CNNs outperforms linear estimators.

The biggest drawback of linear estimators is that they cannot perform feature learning. To investigate this properly rigorously, we parameterize the range of important features and how it affects the estimation accuracy. It is expected that as the range of important features becomes wider, the linear estimator suffers sub-optimality due to their disability of feature learning. To justify this intuition, we introduce the following set of $a$, denoted by $\Gamma$.

**Definition 8.** Let $0 < \epsilon < 1$ and $\eta > 1$ be given and fixed. (i) For mixed smoothness, suppose that $\underline{a}$ and $c$ be constants such that $\underline{a} > \frac{1}{2}$, $0 < c < 2\underline{a} - 1$, and define

$$Q_\epsilon^m := \underline{a}(\underline{a} - 1/2)/\left(c \log_2 \epsilon^{-1}\right). \tag{4}$$

(ii) For anisotropic smoothness, suppose that constants $\underline{a}$ and $c$ satisfy $2\zeta(\eta) < \underline{a}$ and $0 < c < \tilde{a}/2 - 1$ where $\tilde{a} := \underline{a}/\zeta(\eta)$ and $\zeta(\eta) := \sum_{n=1}^\infty n^{-\eta}$. Then, we define

$$Q_\epsilon^a := 2\eta \underline{a} \tilde{a} / \left(\left(\frac{2(1+c)(\eta\tilde{a}+1)}{\tilde{a}} - 1\right) \log_2 \epsilon^{-1} + \underline{a}^{1-1/\eta}\left(\log_2 \epsilon^{-1}\right)^{1/\eta}\right). \tag{5}$$

For $Q = Q_\epsilon^m$ or $Q = Q_\epsilon^a$, we define the set of possible values of $a$ as $\Gamma(Q) := \left\{a : a_i \geq Q \log_2 i, \; \|a\|_{wl^\eta} \leq \underline{a}^{-1}\right\}$.

This definition extends the concept of sparsity for $a$ (see Definition 7). In essence, $\Gamma$ represents a set of $a$ values that exhibit sparsity and are greater than a specific minimum value. We can see that the parameter $c$ controls the minimum value of $a$ through $Q(= Q_\epsilon^m, Q_\epsilon^a)$. Indeed, as $c$ becomes large, the lower bound of $a_i$ becomes smaller, which indicates that the range of coordinates $i$ with small smoothness parameter $a_i$ (in other words, important features) is widely spread out. Such a situation is more difficult for the linear estimators while deep learning approach can adaptively specify important features from wide range of coordinates. This yields separation between deep learning and linear estimators.

Using the $\Gamma$ defined above, we define the union of $\gamma$-smooth spaces over the choice of $a$ in $\Gamma$. By considering such a set as the space containing the true function, it is expected that the feature extraction abilities of dilated CNNs will be demonstrated.

**Definition 9.** Let the set of possible values for $a$ be $\Gamma$, and define $(F_{p,q}(\Gamma))^\infty$ as follows: $(F_{p,q}(\Gamma))^\infty := \bigcup_{a \in \Gamma} \left(\mathcal{F}_{p,q}^{\gamma_a}\right)^\infty$. Also, let $U\left((F_{p,q}(\Gamma))^\infty\right) := \bigcup_{a \in \Gamma} \left(U\left(\mathcal{F}_{p,q}^{\gamma_a}\right)\right)^\infty$.

It is known that the minimax rate in the class of linear estimators is same as that on the *convex hull* of the target function class (Theorem 3.3 of Hayakawa & Suzuki (2020)). Since the union of $\gamma$-smooth spaces over different values of $a$ becomes highly non-convex, it is expected that the linear estimators suffer from sub-optimal rate on the space $(F_{p,q}(\Gamma))^\infty$. Indeed, we obtain the following lower bound of estimation errors for linear estimators.

**Theorem 4** (Estimation error of linear estimators in $(F_{2,2}(\Gamma))^\infty$). Let $\Gamma = \Gamma(Q_\epsilon^m)$ and $a^\star = \underline{a}$ when $\gamma$ is mixed smoothness ($\gamma(s) = \langle a, s \rangle$), and let $\Gamma = \Gamma(Q_\epsilon^a)$ and $a^\star = \tilde{a}$ when $\gamma$ is anisotropic smoothness ($\gamma(s) = \max_i\{a_i s_i\}$). Then, the minimax rate over the class of linear estimators is given as follows:

$$\inf_{\hat{f}:\text{linear}} \sup_{f^\circ \in U((F_{2,2}(\Gamma))^\infty) \cap \mathcal{B}_r} \mathbb{E}_{D_n}\left[\left\|\hat{f} - f^\circ\right\|_{L^2(P_X)}\right] \gtrsim n^{-\frac{2a^\star}{2a^\star+1+c}}.$$

This result achieves a rate that is larger by a factor of $c$, which has not been seen in existing analyses such as for Besov spaces (Donoho & Johnstone, 1998; Zhang et al., 2002). As we have mentioned, this is due to the fact that the linear estimators cannot perform feature learning. More technically, the convex hull of the class $U((F_{2,2}(\Gamma))^\infty)$ become much larger than it as $c$ becomes larger.

**Remark 5.** In the finite dimensional Besov space setting, it has been shown that DNN approach achieves the rate independent of $v$ due to the adaptivity of DNNs to a local smoothness structure while the linear estimator suffers from sub-optimal rate due to the term $v$ (Suzuki, 2019; Suzuki & Nitanda, 2021). However, it is still an open problem whether DNNs can achieve the rate without the term $v$ in our infinite dimensional setting. Then, we employed a different strategy to show the separation between the dilated CNNs and linear estimators, that is, we considered a larger function class by controlling $\Gamma$.

As in Theorem 3, we can easily show the following upper bound for the extended class $(F_{2,2}(\Gamma))^\infty$.

**Corollary 6** (Estimation error of extended CNNs for $(F_{2,2}(\Gamma))^\infty$). In accordance with the definition used in Theorem 4, we define and utilize $\Gamma, \gamma_a(s)$, and $a^\star$. Also, if the true nonlinear operator $f^\circ$ satisfies $f^\circ \in U((F_{2,2}(\Gamma))^\infty)$ and Eq (3), the estimation error of dilated CNNs is given as follows:

$$\mathbb{E}_{P^n}\left[\left\|\hat{f} - f^\circ\right\|_{P_X}^2\right] \lesssim n^{-\frac{(2-r)a^\star}{2a^\star+1}} (\log n)^{\frac{2}{\eta}+6}.$$

From the above corollary, we can compare with the estimation error of linear estimators obtained in Theorem 4. The following theorem states this comparison.

**Theorem 7** (Superiority of dilated CNNs over linear estimators). Assuming the same conditions as in Corollary 6, when condition $c > \frac{(2a^\star+1)r}{2-r}$ is satisfied, dilated CNNs outperform linear estimators. However, note that it is necessary for $r < \frac{2a^\star-1}{2a^\star}$, $a^\star > \frac{1}{2}$ to exist such $c$ for $\Gamma = \Gamma(Q_\epsilon^m)$, $\gamma_a(s) = \langle a, s \rangle$ and $a^\star = \underline{a}$. Also, it is necessary for $r < \frac{2}{5} \cdot \frac{a^\star-2}{a^\star}$, $a^\star > 2$ to exist such $c$ for $\Gamma = \Gamma(Q_\epsilon^a)$, $\gamma_a(s) = \max_i\{a_i s_i\}$, and $a^\star = \tilde{a}$.

In previous studies, the superiority of deep learning over linear estimators, among others, has been attributed to the adaptivity of deep learning to spatial inhomogeneity (Imaizumi & Fukumizu (2019); Suzuki (2019)). However, as mentioned at the beginning of Section 4.1, the approximation error of dilated CNNs is derived using a non-adaptive approach in this work. Even in such cases, demonstrating the superiority by leveraging the feature extraction abilities of dilated CNNs represents a novel perspective to highlight the advantages of deep learning.

## 5 CONCLUSION

In this study, we investigated the performance of dilated CNNs in infinite-dimensional input and output spaces. Our primary findings revealed that the convergence rates of dilated CNNs depend solely on the smoothness and decay rate of the output, when considering nonlinear operators composed of a countably infinite sequence of functions belonging to the $\gamma$-smooth space, and the convergence rate achieves minimax optimality. Additionally, we showed that dilated CNNs outperform linear methods in cases where the smoothness of the $\gamma$-smooth space varies, emphasizing the advantages of neural network-based feature extraction. These results provide a theoretical basis for the success of deep learning in high-dimensional input-output tasks and highlight the potential of dilated CNNs in handling complex data. As for future work, several directions can be pursued, including: (i) demonstrating the rate of approximation errors for dilated CNNs using adaptive methods, and (ii) extending the analysis to models other than dilated CNNs, exploring their performance in similar infinite-dimensional input and output settings.

ACKNOWLEDGEMENT

YN was partially supported by JST CREST (JPMJCR2015). TS was partially supported by JSPS KAKENHI (20H00576) and JST CREST (JPMJCR2115).

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

# A    PROOFS OF THEOREM 1 AND 3

Here, we present the proof of the theorem that provides an approximation error and estimation error using dilated CNNs. These proofs are essentially an extension of the results in Okumoto & Suzuki (2021), which demonstrate the approximation error and estimation error in one-dimensional output, to the case of infinite-dimensional output.

## A.1    PROOF OF THEOREM 1

We begin by showing the proof of Theorem 1, which demonstrates the approximation error of dilated CNNs. Before delving into the main topic, we introduce several symbols and provide their definitions.

**Definition 10** (Axial complexity and frequency direction complexity). Let $T > 0$ be a constant and $\gamma : \mathbb{N}_0^\infty \to \mathbb{R}_{>0}$ be a smoothness function. We define the set

$$I(T, \gamma) := \{i \in \mathbb{N} : \exists s \in \mathbb{N}_0^\infty, \ s_i \neq 0, \ \gamma(s) < T\},$$

and we define the **axial complexity** for $I(T, \gamma)$ as follows:

$$d_{\max}(T, \gamma) := |I(T, \gamma)|.$$

Furthermore, the **frequency direction complexity** is defined as:

$$f_{\max}(T, \gamma) := \max_{s \in \mathbb{N}_0^\infty : \gamma(s) < T} \ \max_{i \in \mathbb{N}} s_i.$$

The axial complexity represents the number of necessary inputs when evaluating the restricted infinite-dimensional space where $\gamma(s) < T$. Similarly, the frequency direction complexity indicates how far the evaluation needs to extend in terms of frequency when the space is restricted.

Furthermore, we introduce the following symbols:

$$v := \left(\frac{1}{p} - \frac{1}{2}\right)_+, \ \alpha(\gamma) := \sup_{s \in \mathbb{N}_0^\infty} \frac{\sum_{i=1}^\infty s_i}{\gamma(s)}, \ G(T, \gamma) := \sum_{s \in \mathbb{N}_0^\infty : \gamma(s) < T} 2^s,$$

where $(x)_+ = \max\{x, 0\}$.

Now, in order to provide a proof for Theorem 1, we show the following theorem:

**Theorem 8** (Approximation error of FNNs). Let $\gamma, \gamma' : \mathbb{N}_0^\infty \to \mathbb{R}_{>0}$ satisfy

$$\gamma'(s) < \gamma(s), \ v\alpha(\gamma) < 1, \ v\alpha(\gamma') < 1$$

and assume that the true nonlinear operator $f^\circ \in \left(\mathcal{F}_{p,q}^\gamma\right)^\infty$ satisfies Assumption 3. For any $T > 0$, define $(d_{\max}, f_{\max}, G)$ as follows:

$$(d_{\max}, f_{\max}, G) = \begin{cases} (d_{\max}(T, \gamma), f_{\max}(T, \gamma), G(T, \gamma)) & (1 \leq q \leq 2), \\ (d_{\max}(T, \gamma'), f_{\max}(T, \gamma'), G(T, \gamma')) & (2 < q). \end{cases}$$

And for constants $K, K' > 0$ depending on $B_2$ and $r$, we set:

$$L = 2K \max\left\{d_{\max}^2, T^2, (\log G)^2, \log f_{\max}\right\}, \ W = \max\left\{21 d_{\max} G, \ (B_2)^r 2^{r(1-v\alpha)T}\right\},$$

$$S = 1764 K (B_2)^r d_{\max}^2 \max\left\{d_{\max}^2, T^2, (\log G)^2, \log f_{\max}, 2^{r(1-v\alpha)T}\right\} G, \ B = \left(\sqrt{2}\right)^{d_{\max}} K'.$$

In this case, there exists an FNNs $\hat{R}_T \in \Phi(L, W, S, B)$ that takes $d_{\max}$-dimensional inputs and returns $d_{L+1}$-dimensional outputs, and for $x = (x_i)_{i=1}^\infty \in \mathcal{M}$, we define $f'(x) : \mathcal{M} \to \mathbb{R}^\infty$ as follows:

$$(f'(x))_i = \begin{cases} \left(\hat{R}_T\left((x_i)_{i \in I(T, \gamma)}\right)\right)_i & (1 \leq i \leq d_{L+1}), \\ 0 & (d_{L+1} < i), \end{cases}$$

where $d_{L+1} = \lfloor (B_2)^r 2^{r(1-v\alpha)T} \rfloor$. Then, we obtain the following inequality:

$$\|f' - f^\circ\|_2 \lesssim \begin{cases} 2^{-\left(1-\frac{r}{2}\right)(1-v\alpha(\gamma))T} & (1 \leq q \leq 2), \\ 2^{-\left(1-\frac{r}{2}\right)(1-v\alpha(\gamma'))T} \left(\sum_{T \leq \gamma'(s)} 2^{\frac{2q}{q-2}(\gamma'(s) - \gamma(s))}\right)^{1/2 - 1/q} & (2 < q). \end{cases}$$

The proof of this theorem is given in A.1.1.

Note that, in Theorem 1, the symbols $L'$, $B'$, $W'$, $C$, $a^\dagger$ should be defined depending on the conditions on $a$ and $\gamma$. Indeed, remember that there are two different settings regarding to $a$: (i) polynomial-order-increasing setting, and (ii) sparse setting; and as for $\gamma$, there are two settings: (i) mixed smoothness and (ii) anisotropic smoothness. Then, each quantity should have different definition according to the combination of the conditions on $a$ and $\gamma$.

However, it is worth noting that the proof itself is nearly identical to that presented in Okumoto & Suzuki (2021), with the only difference being the use of Theorem 8 instead of Theorem 7 in Okumoto & Suzuki (2021). Therefore, in this paper, we provide only the proof for the case where $a$ increases in polynomial order and $\gamma$ is of mixed smoothness. The same argument is also applied to the other settings.

*Proof of Theorem 1.* (In the case where $a$ increases in polynomial order and $\gamma$ is of mixed smoothness). First, we discuss the case of $1 \leq q \leq 2$. According to Lemma 18 in Okumoto & Suzuki (2021), we obtain the following bound:

$$G\left(T, \gamma\right) = \sum_{s \in \mathbb{N}_0^\infty : \gamma(s) < T} 2^s = \sum_{s \in \mathbb{N}_0^\infty : \langle \frac{a}{a_1}, s \rangle < \frac{T}{a_1}} 2^s \leq 8 \left( \prod_{i=2}^\infty \frac{1}{1 - 2^{\frac{-(a_i - a_1)}{a_1}}} \right) 2^{\frac{T}{a_1}}$$

Furthermore, since $a$ is monotonically increasing and $a_i = \Omega(i^\eta)$, we obtain:

$$\alpha = \sup_{s \in \mathbb{N}_0^\infty} \frac{\sum_{i=1}^\infty s_i}{\langle a, s \rangle} = \frac{1}{a_1}, \ d_{\max} \sim T^{\frac{1}{\eta}}, \ f_{\max} \sim T.$$

Now, by using the filter $w \in \mathbb{R}^{C \times 1 \times W'}$ with the width $W' = d_{\max}$, the number of output channels $C = d_{\max}$ and the number of input channels $C' = 1$ given by

$$w_{i,1,j} = \begin{cases} 1 & (i = j), \\ 0 & (i \neq j), \end{cases}$$

for $i, j \in [d_{\max}]$, we can see that

$$\left(\mathrm{Conv}_{1,w}\left(X\right)\right)_1 = \begin{pmatrix} x_1 \\ \vdots \\ x_{d_{\max}} \end{pmatrix}.$$

By Theorem 8, if we set

$$L = 2K \max \left\{ T^{\frac{2}{\eta}}, \ T^2 \right\},$$

$$W = \max \left\{ 21 \left( \prod_{i=2}^\infty \frac{1}{1 - 2^{\frac{-(a_i - a_1)}{a_1}}} \right) T^{\frac{1}{\eta}} 2^{\frac{T}{a_1}}, \ (B_2)^r 2^{r\left(1 - \frac{v}{a_1}\right)T} \right\},$$

$$S = 1764K (B_2)^r \left( \prod_{i=2}^\infty \frac{1}{1 - 2^{\frac{-(a_i - a_1)}{a_1}}} \right) T^{\frac{2}{\eta}} 2^{r\left(1 - \frac{v}{a_1}\right)T} 2^{\frac{T}{a_1}},$$

$$B = \left( \sqrt{2} \right)^{T^{\frac{1}{\eta}}} K',$$

where $K, K' > 0$ are constants, for a true function $f^\circ \in \left( U\left(\mathcal{F}_{p,q}^\gamma\right) \right)^\infty$ satisfying Assumption 3, there exists an FNNs $\hat{R}_T \in \Phi\left(L, W, S, B\right)$ such that

$$\left(f''\left(X\right)\right)_i = \begin{cases} \left( \hat{R}_T \left(\mathrm{Conv}_{1,w}\left(X\right)\right)_1 \right)_i & (1 \leq i \leq d_{L+1}), \\ 0 & (d_{L+1} < i), \end{cases}$$

where $d_{L+1} = \lfloor (B_2)^r 2^{r(1 - v\alpha)T} \rfloor$, satisfies

$$\|f'' - f^\circ\|_2 \lesssim 2^{-\left(1 - \frac{r}{2}\right)\left(1 - \frac{v}{a_1}\right)T}.$$

Here, by defining $(f'(X))_i := -B_2 i^{-\frac{1}{r}} \vee \left( B_2 i^{-\frac{1}{r}} \wedge (f''(X))_i \right)$ $(\forall i \in \mathbb{N})$, $f'$ can be represented as a dilated CNNs $f' \in \bar{\mathcal{P}}(B_2, L', B', W', C, L, W, S, B)$, where $L' = 1$, $B' = 1$, and $W' = C = d_{\max}$.

Furthermore, from Assumption 3, we obtain a following relationship:

$$\|f' - f^\circ\|_2 \leq \|f'' - f^\circ\|_2 \lesssim 2^{-\left(1-\frac{r}{2}\right)\left(1-\frac{v}{a_1}\right)T}.$$

Thus, the theorem's statement is proven.

Next, we consider the case of $q > 2$. In this case, let $a_1' = \frac{a_1}{2}$, $\delta = a_2 - a_1$ (where $\delta > 0$ by definition of $a$), and a constant $u$ satisfying $2 < u < 2 + \frac{2\delta}{a_1}$. For $i \geq 1$, we define $a_i' = \frac{a_i}{u}$. Then, as shown in the proof of Theorem 9 in Okumoto & Suzuki (2021), we have:

$$\left( \sum_{T \leq \gamma'(s)} 2^{\frac{2q}{q-2}\left(\gamma'(s)-\gamma(s)\right)} \right)^{1/2-1/q} \leq \left( 2^{-\frac{2q}{q-2}T} \right)^{1/2-1/q} = 2^{-T} \leq 2^{-\left(1-\frac{r}{2}\right)T}.$$

Therefore, similarly to the case of $1 \leq q \leq 2$, using Theorem 8, we can conclude that for a true function $f^\circ$ satisfying Assumption 3, there exists $f' \in \bar{\mathcal{P}}$ such that:

$$\|f' - f^\circ\|_2 \lesssim 2^{-\left(1-\frac{r}{2}\right)(1-v\alpha)T} \left( \sum_{T \leq \gamma'(s)} 2^{\frac{2q}{q-2}\left(\gamma'(s)-\gamma(s)\right)} \right)^{1/2-1/q} \leq 2^{-2\left(1-\frac{r}{2}\right)\left(1-\frac{v}{a_1}\right)T},$$

where $\alpha(\gamma') = \frac{2}{a_1}$. Now, using Lemma 18 from Okumoto & Suzuki (2021) again, we have:

$$G(T, \gamma') \leq 8 \left( \prod_{i=2}^{\infty} \frac{1}{1 - 2^{\frac{-(a_i'-a_1')}{a_1'}}} \right) 2^{\frac{T}{a_1'}} \leq 8 \left( \prod_{i=2}^{\infty} \frac{1}{1 - 2^{\frac{-(a_i-a_1)}{a_1}}} \right) 2^{\frac{2T}{a_1}}.$$

Therefore, by setting $T \leftarrow 2T$, we can establish the result for the case of $1 \leq q \leq 2$. $\blacksquare$

### A.1.1 PROOF OF THEOREM 8

To prove this theorem, we first introduce the following function as an approximation of $\tilde{f} \in \mathcal{F}_{p,q}^\gamma$:

$$\tilde{R}_T(\tilde{f}) := \begin{cases} \sum_{s \in \mathbb{N}_0^\infty : \gamma(s) < T} \delta_s(\tilde{f}) & (1 \leq q \leq 2), \\ \sum_{s \in \mathbb{N}_0^\infty : \gamma'(s) < T} \delta_s(\tilde{f}) & (2 < q). \end{cases}$$

Based on this notation, we also define $R_T(f) : \mathcal{M} \to \mathbb{R}^\infty$ for any $d_{\text{out}} \in \mathbb{N}$ as an approximation of $f \in \left(\mathcal{F}_{p,q}^\gamma\right)^\infty$ as

$$(R_T(f)(x))_i = \begin{cases} \tilde{R}_T(f_i)(x) & (1 \leq i \leq d_{\text{out}}), \\ 0 & (d_{\text{out}} < i). \end{cases}$$

To establish the proof of Theorem 8, we present the following lemma.

**Lemma 9.** Assume that $\gamma, \gamma' : \mathbb{N}_0^\infty \to \mathbb{R}_{>0}$ satisfy:

$$\gamma'(s) < \gamma(s), \ v\alpha(\gamma) < 1, \ v\alpha(\gamma') < 1,$$

and a true nonlinear operator $f \in \left(\mathcal{F}_{p,q}^\gamma\right)^\infty$ satisfies Assumption 3.

Under these conditions, we obtain the following inequality:
$\underline{1 \leq q \leq 2}$

$$\|f - R_T(f)\|_2 \leq 2^{-(1-v\alpha(\gamma))T} \sqrt{d_{\text{out}}} + B_2 \sqrt{\frac{r}{2-r}} d_{\text{out}}^{\frac{1}{2}-\frac{1}{r}}.$$

$\underline{2 < q}$

$$\|f - R_T(f)\|_2 \leq 2^{-\left(1-v\alpha(\gamma')\right)T} \left[ \sum_{T \leq \gamma'(s)} 2^{\frac{2q}{q-2}\left(\gamma'(s)-\gamma(s)\right)} \right]^{1-2/q} \sqrt{d_{\text{out}}} + B_2 \sqrt{\frac{r}{2-r}} d_{\text{out}}^{\frac{1}{2}-\frac{1}{r}}.$$

This proof is provided in A.1.2.

Now, using this lemma, we proceed with the proof of Theorem 8. The strategy is as follows: since Lemma 9 gives us the error when approximating the true function $f^\circ$ by $R_T(f^\circ)$, we aim to obtain the approximation error when approximating $R_T(f^\circ)$ using FNNs. By summing up these errors, we can obtain the approximation error when approximating $f^\circ$ using FNNs.

*Proof of Theorem 8.* Now, according to Theorem 4.1 in Perekrestenko et al. (2018), for any $\epsilon > 0$ and some constants $C_1, C_2 > 0$, if we let

$$L_{\tilde{\psi}} = C_1 \left[ \left( \log \frac{1}{\epsilon} \right)^2 + \log(f_{\max}) \right],$$

there exists a neural network $\tilde{\psi}_{l_i} \in \Phi(L_{\tilde{\psi}}, 21, C_2, 21^2 L_{\tilde{\psi}})$ that approximates $\psi_{l_i}$ with the following accuracy:

$$\left\| \psi_{l_i} - \tilde{\psi}_{l_i} \right\|_{L^\infty([0,1])} \le \epsilon.$$

For such $\tilde{\psi}_{l_i}$, we define:

$$\hat{\psi}_{l_i} = \max \left\{ -\sqrt{2}, \min \left\{ \sqrt{2}, \tilde{\psi}_{l_i} \right\} \right\}, L_{\tilde{\psi}_{l_i}} = C_1 \left[ \left( \log \frac{1}{\epsilon} \right)^2 + \log(f_{\max}) \right] + 2.$$

Moreover, according to Proposition 3 in D.Yarotsky (2017), for any $\epsilon > 0$ and a constant $B_\times > 0$, if we let

$$L_\times = \left\lceil \log \left( \frac{3^{d_{\max}}}{\epsilon} + 5 \right) \right\rceil \lceil \log d_{\max} \rceil, W_\times = 6 d_{\max}, S_\times = L_\times W_\times^2,$$

there exists a neural network $\phi_\times \in \Phi(L_\times, W_\times, B_\times, S_\times)$ satisfying:

$$\left\| \phi_\times - \prod_{i=1}^{d_{\max}} x_i \right\|_{L^\infty([-1,1]^{d_{\max}})} \le \epsilon.$$

Now, according to the proof of Theorem 7 in Okumoto & Suzuki (2021), for $f_i \in \mathcal{F}_{p,q}^\gamma$, we define:

$$\hat{R}_T(f_i) := \sum_{\gamma(s) < T} \sum_{l \in J(s)} \left( \sqrt{2} \right)^{d_{\max}} \langle f_i, \psi_l \rangle \phi_\times \left( \frac{\hat{\psi}_{l_1}}{\sqrt{2}}, \dots, \frac{\hat{\psi}_{l_{d_{\max}}}}{\sqrt{2}} \right),$$

where $J(s) := \left\{ l \in \mathbb{Z}_0^\infty : \lfloor 2^{s_i-1} \rfloor \le |l_i| < 2^{s_i} \right\}$. Then, we obtain that

$$\left\| \hat{R}_T(f_i) - R_T(f_i) \right\|_{L^\infty([0,1]^{d_{\max}})} \le B_2 i^{-\frac{1}{r}} G(T, \gamma) \left( \sqrt{2} \right)^{d_{\max}} (d_{\max} + 1) \epsilon,$$

where we used the fact $\langle f_i, \psi_l \rangle \le \|f_i\|_2 \le B_2 i^{-\frac{1}{r}}$. Therefore, as a approximation of $R_T(f)$, for any $d_{\text{out}} \in \mathbb{N}$, we define $\hat{R}_T(f) : [0,1]^{d_{\max}} \to \mathbb{R}^{d_{\text{out}}}$ as follows:

$$\forall i \in [d_{\text{out}}] : \left( \hat{R}_T(f) \right)_i := \hat{R}_T(f_i).$$

This yields a neural network with the same intermediate layer $\phi_\times \left( \frac{\hat{\psi}_{l_1}}{\sqrt{2}}, \dots, \frac{\hat{\psi}_{l_{d_{\max}}}}{\sqrt{2}} \right)$, but with an output dimension increased from one to $d_{\text{out}} = d_{L+1}$.

Then, by using this $\hat{R}_T(f)$, for $x = (x_i)_{i=1}^\infty \in \mathcal{M}$, we define $f'(x) : \mathcal{M} \to \mathbb{R}^\infty$ as follows:

$$(f'(x))_i = \begin{cases} \left( \hat{R}_T \left( (x_i)_{i \in I(T,\gamma)} \right) \right)_i & (1 \le i \le d_{\text{out}}), \\ 0 & (d_{\text{out}} < i). \end{cases}$$

With this $f'$, the approximation of $R_T(f)$ results in the following error:

$$\|R_T(f) - f'\|_{L^\infty\left([0,1]^{d_{\max}}\right)} \leq \sum_{i=1}^{d_{\text{out}}} B_2 i^{-\frac{1}{r}} G(T,\gamma) \left(\sqrt{2}\right)^{d_{\max}} (d_{\max}+1)\epsilon$$

$$\leq d_{\text{out}} B_2 G(T,\gamma) \left(\sqrt{2}\right)^{d_{\max}} (d_{\max}+1)\epsilon.$$

Therefore, in the case of $1 \leq q \leq 2$, with the result of Lemma 9, we get the following inequality:

$$\|f - f'\|_2 \leq \|f - R_T(f)\|_2 + \|R_T(f) - f'\|_{L^\infty\left([0,1]^{d_{\max}}\right)}$$

$$\leq 2^{-(1-v\alpha)T}\sqrt{d_{\text{out}}} + B_2\sqrt{\frac{r}{2-r}} d_{\text{out}}^{\frac{1}{2}-\frac{1}{r}} + d_{\text{out}} B_2 G(T,\gamma) \left(\sqrt{2}\right)^{d_{\max}} (d_{\max}+1)\epsilon.$$

Here, we put

$$\epsilon = \frac{2^{-(1-v\alpha)T}\sqrt{d_{\text{out}}} - B_2\sqrt{\frac{r}{2-r}} d_{\text{out}}^{\frac{1}{2}-\frac{1}{r}}}{d_{\text{out}} B_2 G(T,\gamma) \left(\sqrt{2}\right)^{d_{\max}} (d_{\max}+1)}.$$

From the condition of $\epsilon > 0$, $d_{\text{out}}$ need to satisfy the following inequality:

$$2^{-(1-v\alpha)T}\sqrt{d_{\text{out}}} - B_2\sqrt{\frac{r}{2-r}} d_{\text{out}}^{\frac{1}{2}-\frac{1}{r}} > 0 \iff d_{\text{out}} > \left(B_2\sqrt{\frac{r}{2-r}}\right)^r 2^{r(1-v\alpha)T}$$

Now, since $0 < r < 1$, we have $0 < \frac{r}{2-r} < 1$. Then, we define $d_{\text{out}}$ as

$$d_{\text{out}} = (B_2)^r 2^{r(1-v\alpha)T}.$$

Then, we can obtain the following inequality:

$$\|f - f'\|_2 \lesssim 2^{-(1-v\alpha)T}\sqrt{d_{\text{out}}} \lesssim 2^{-\left(1-\frac{r}{2}\right)(1-v\alpha)T}.$$

The case of $2 < q$ can be proved in the same manner.

Finally, we evaluate the size of the neural network. As mentioned earlier, since $R_T(f)$ is a linear combination of the neural network $\phi_\times\left(\frac{\hat{\psi}_{l_1}}{\sqrt{2}}, \ldots, \frac{\hat{\psi}_{l_{d_{\max}}}}{\sqrt{2}}\right)$, if we let

$$L = L_{\hat{\phi}} + L_\times + 1,$$
$$W = \max\{21 d_{\max} G(T,\gamma),\ d_{\text{out}}\},$$
$$S = \left(21^2 d_{\max} L_{\hat{\phi}} + L_\times W_\times^2 + d_{\text{out}}\right) G(T,\gamma),$$
$$B = \max\left\{C_2,\ B_\times,\ \left(\sqrt{2}\right)^{d_{\max}} B_f\right\},$$

$\hat{R}_T(f)$ satisfies $\hat{R}_T(f) \in \Phi(L,W,S,B)$.

First, we evaluate $L_{\hat{\phi}}$ as follows:

$$L_{\hat{\phi}} = C_1 \left\lfloor \left(\left(\log\frac{1}{\epsilon}\right)^2 + \log(f_{\max})\right) + 2 \right\rfloor$$

$$= C_1 \left\lfloor \left(\frac{r}{2}\log B_2 + r(1-v\alpha)T\log 2 + \log B_2 + \log G(T,\gamma) + \frac{d_{\max}}{2}\log 2 \right.\right.$$

$$\left.\left. + \log(d_{\max}+1) + \left(1-\frac{r}{2}\right)(1-v\alpha)T\log 2 + \log\frac{1}{1-\sqrt{\frac{r}{2-r}}}\right)^2 + \log(f_{\max}) + 2 \right\rfloor.$$

Now, considering that $0 < r < 1$, we note that:

$$\frac{1}{1-\sqrt{\frac{r}{2-r}}} = \frac{2-r+\sqrt{r(2-r)}}{2(1-r)} \leq 1 + \frac{1}{1-r} \leq \frac{2}{1-r}.$$

Using this, we can obtain the following upper bound for $L_{\hat{\phi}}$:

$$L_{\hat{\phi}} \leq \left\lfloor C_1 \left( 7 \max \left\{ \log B_2, \ \log \frac{2}{1-r} \right\} \right)^2 \right\rfloor \left\lfloor \max \left\{ d_{\max}^2, \ T^2, \ (\log G(T,\gamma))^2, \ \log f_{\max}, \ 2 \right\} \right\rfloor$$

Similarly, we can evaluate $L_\times$ as:

$$L_\times \leq \left\lfloor 7 \max \left\{ \log B_2, \ \log \frac{2}{1-r}, \ \log 5 \right\} \right\rfloor \left\lfloor \max \left\{ d_{\max}, \ T, \ \log G(T,\gamma) \right\} \right\rfloor \left\lfloor \log d_{\max} \right\rfloor.$$

Then, if we let

$$K = 2 \max \left\{ \left\lfloor C_1 \left( 7 \max \left\{ \log B_2, \ \log \frac{2}{1-r} \right\} \right)^2 \right\rfloor, \ \left\lfloor 7 \max \left\{ \log B_2, \ \log \frac{2}{1-r}, \ \log 5 \right\} \right\rfloor \right\},$$

we can obtain the following upper bound for $L$:

$$L \leq 2K \left\lfloor \max \left\{ d_{\max}^2, \ T^2, \ (\log G(T,\gamma))^2, \ \log f_{\max} \right\} \right\rfloor.$$

Similarly, we can evaluate $S$ as:

$$S = \left( 21^2 d_{\max} L_{\hat{\phi}} + L_\times W_\times^2 + d_{\mathrm{out}} \right) G(T,\gamma)$$

$$\leq \max \left\{ 4 \times 21^2 K d_{\max}^2 \max \left\{ d_{\max}^2, \ T^2, \ (\log G(T,\gamma))^2, \ \log f_{\max} \right\}, \ (B_2)^r \, 2^{r(1-v\alpha)T} \right\} G(T,\gamma)$$

$$\leq 4 \times 21^2 K (B_2)^r \, d_{\max}^2 \max \left\{ d_{\max}^2, \ T^2, \ (\log G(T,\gamma))^2, \ \log f_{\max}, \ 2^{r(1-v\alpha)T} \right\} G(T,\gamma).$$

∎

### A.1.2 PROOF OF LEMMA 9

*Proof of Lemma 9.* Now,

$$\|f - R_T(f)\|_2^2 = \int_{\mathcal{M}} \|f(x) - R_T(f)(x)\|_{\ell^2}^2 \, \mathrm{d}\lambda^\infty(x)$$

$$= \int_{\mathcal{M}} \sum_{i=1}^\infty (f_i(x) - (R_T(f)(x))_i)^2 \, \mathrm{d}\lambda^\infty(x)$$

$$= \int_{\mathcal{M}} \left( \sum_{i=1}^{d_{\mathrm{out}}} \left( f_i(x) - \tilde{R}_T(f_i)(x) \right)^2 + \sum_{i=d_{\mathrm{out}}+1}^\infty f_i(x)^2 \right) \mathrm{d}\lambda^\infty(x)$$

$$\leq \sum_{i=1}^{d_{\mathrm{out}}} \left\| f_i - \tilde{R}_T(f_i) \right\|_2^2 + (B_2)^2 \sum_{i=d_{\mathrm{out}}+1}^\infty i^{-\frac{2}{r}}$$

holds. Here, the last inequality is derived using Eq (3). Now, with regard to $\left\| f_i - \tilde{R}_T(f_i) \right\|_2^2$, Lemma 17 from Okumoto & Suzuki (2021) states that, for example, when $1 \leq q \leq 2$,

$$\left\| f_i - \tilde{R}_T(f_i) \right\|_2^2 \leq 2^{-2(1-v\alpha)T} \|f_i\|_{\mathcal{F}_{p,q}^\gamma}^2$$

is valid. Combining this with Eq (2) in Assumption 3, we have

$$\sum_{i=1}^{d_{\mathrm{out}}} \left\| f_i - \tilde{R}_T(f_i) \right\|_2^2 \leq \sum_{i=1}^{d_{\mathrm{out}}} 2^{-2(1-v\alpha)T} = 2^{-2(1-v\alpha)T} d_{\mathrm{out}}.$$

This is also true for $q > 2$. Additionally, in general,

$$\sum_{i=m+1}^\infty i^{-s} \leq \int_m^\infty \frac{1}{x^s} \mathrm{d}x = \frac{m^{1-s}}{s-1}.$$

can be obtained through simple integration calculations. Thus,

$$\sum_{i=d_{\text{out}}+1}^{\infty} i^{-\frac{2}{r}} \leq \frac{r}{2-r} d_{\text{out}}^{1-\frac{2}{r}}$$

is obtained. Hence, in the case of $1 \leq q \leq 2$, we obtain the following inequalities:

$$\|f - R_T(f)\|_2^2 \leq 2^{-2(1-v\alpha)T} d_{\text{out}} + (B_2)^2 \frac{r}{2-r} d_{\text{out}}^{1-\frac{2}{r}}.$$

$$\therefore \|f - R_T(f)\|_2 \leq \left( 2^{-2(1-v\alpha)T} d_{\text{out}} + (B_2)^2 \frac{r}{2-r} d_{\text{out}}^{1-\frac{2}{r}} \right)^{\frac{1}{2}}$$

$$\leq 2^{-(1-v\alpha)T} \sqrt{d_{\text{out}}} + B_2 \sqrt{\frac{r}{2-r}} d_{\text{out}}^{\frac{1}{2}-\frac{1}{r}}.$$

The desired inequality can be obtained in a similar manner for the case when $q > 2$. ∎

## A.2 PROOF OF THEOREM 2

*Proof of Theorem 2.* First, $\mathbb{E}_{P^n} \left[ \|\hat{f} - f^\circ\|_{P_X}^2 \right]$ can be decomposed into two partial sums as follows:

$$\mathbb{E}_{P^n} \left[ \|\hat{f} - f^\circ\|_{P_X}^2 \right] = \mathbb{E}_{P^n} \left[ \mathbb{E}_{P_X} \left[ \|\hat{f}(X) - f^\circ(X)\|_{\ell^2}^2 \right] \right]$$

$$= \mathbb{E}_{P^n} \left[ \mathbb{E}_{P_X} \left[ \sum_{i=1}^{\infty} \left( \hat{f}_i(X) - f_i^\circ(X) \right)^2 \right] \right]$$

$$= \sum_{i=1}^{\infty} \mathbb{E}_{P^n} \left[ \mathbb{E}_{P_X} \left[ \left( \hat{f}_i(X) - f_i^\circ(X) \right)^2 \right] \right]$$

$$= \sum_{i=1}^{\infty} \mathbb{E}_{P^n} \left[ \|\hat{f}_i - f_i^\circ\|_{P_X}^2 \right]$$

$$\lesssim \sum_{i=1}^{d_{\text{out}}} \mathbb{E}_{P^n} \left[ \|\hat{f}_i - f_i^\circ\|_{P_X}^2 \right] + \sum_{d_{\text{out}}+1}^{\infty} \mathbb{E}_{P^n} \left[ \|\hat{f}_i - f_i^\circ\|_2^2 \right],$$

where we utilize the assumption of the Radon-Nikodym derivative, which implies $\|\cdot\|_{P_X} \lesssim \|\cdot\|_2$.

First, for the second term, according to Assumption 3, we can evaluate it as follows:

$$\sum_{d_{\text{out}}+1}^{\infty} \mathbb{E}_{P^n} \left[ \|\hat{f}_i - f_i^\circ\|_2^2 \right] \leq \sum_{d_{\text{out}}+1}^{\infty} \mathbb{E}_{P^n} \left[ 4(B_2)^2 i^{-\frac{2}{r}} \right] \leq \frac{4r}{2-r} (B_2)^2 d_{\text{out}}^{1-\frac{2}{r}}.$$

This provides an upper bound for the second term.

For the first term, we can utilize the results from Theorem 10 and 14 in Okumoto & Suzuki (2021). Thus, we have:

$$\sum_{i=1}^{d_{\text{out}}} \mathbb{E}_{P^n} \left[ \|\hat{f}_i - f_i^\circ\|_{P_X}^2 \right] \lesssim \sum_{i=1}^{d_{\text{out}}} n^{-\frac{2(a^\dagger - v)}{2(a^\dagger - v)+1}} (\log n)^{\frac{2}{\eta}+2} \max \left\{ (\log n)^{\frac{4}{\eta}}, (\log n)^4 \right\}$$

$$\lesssim d_{\text{out}} n^{-\frac{2(a^\dagger - v)}{2(a^\dagger - v)+1}} (\log n)^{\frac{2}{\eta}+2} \max \left\{ (\log n)^{\frac{4}{\eta}}, (\log n)^4 \right\}.$$

Here, Okumoto & Suzuki (2021) choose $T$ such that $T = \frac{a^\dagger}{2(a^\dagger - v)+1} \log_2 n$. Furthermore, from Theorem 1, we have $d_{\text{out}} = (B_2)^2 2^{r\left(1-\frac{v}{a^\dagger}\right)T}$. Substituting these values into the above equation, we obtain:

$$\mathbb{E}_{P^n} \left[ \|\hat{f} - f^\circ\|_{P_X}^2 \right] \lesssim n^{-\frac{(2-r)(a^\dagger - v)}{2(a^\dagger - v)+1}} (\log n)^{\frac{2}{\eta}+2} \max \left\{ (\log n)^{\frac{4}{\eta}}, (\log n)^4 \right\}.$$

∎

# B  PROOF OF THEOREM 2

To prove this theorem, we first define several concepts. First, consider a totally bounded metric space $(\mathcal{F}, \rho)$ consisting of a set $\mathcal{F}$ and a metric $\rho : \mathcal{F} \times \mathcal{F} \to \mathbb{R}_+$. We define a set in $\mathcal{F}$ that satisfies the following condition: for any $\epsilon > 0$, there exists a collection of functions $\{f^1, \ldots, f^N\}$ such that for any $f \in \mathcal{F}$, there exists $k \in [N]$ satisfying $\rho(f, f^k) \le \epsilon$. We refer to such a collection as an $\epsilon$-covering set of $\mathcal{F}$, and the $\epsilon$-covering number $\mathcal{N}(\epsilon, \mathcal{F}, \rho)$ is defined as the size of the smallest $\epsilon$-covering set.

Furthermore, an $\epsilon$-packing set of $\mathcal{F}$ is a collection of functions $\{f^1, \ldots, f^M\} \subset \mathcal{F}$ such that $\rho(f^i, f^j) \ge \epsilon$ for all $i \ne j$. The $\epsilon$-packing number $\mathcal{M}(\epsilon, \mathcal{F}, \rho)$ is defined as the size of the largest $\epsilon$-packing set.

Now, with the definition of the $\epsilon$-packing number, we can establish the following lemma.

**Lemma 10.** Consider $\left(U\left(\mathcal{F}_{p,q}^\gamma\right)\right)^d \subset \left(U\left(\mathcal{F}_{p,q}^\gamma\right)\right)^\infty$ for any $d \ge 4$. In accordance with the definition used in Theorem 1, we define and utilize $a^\dagger$. For any $f \in \left(U\left(\mathcal{F}_{p,q}^\gamma\right)\right)^d$ subject to the constraint $\|f_i\|_2 \le B_2 i^{-\frac{1}{r}}$ $(\forall i \in [d])$, the following inequality holds for any $s \le \frac{d}{4}$, $p \ge 2$:

$$\log \mathcal{M}\left(\delta, \left(U\left(\mathcal{F}_{p,q}^\gamma\right)\right)^d, \|\cdot\|_2\right) \gtrsim s \log \frac{d}{s} + s d^{\frac{1}{ra^\dagger}} \log\left(\frac{d^{-\frac{1}{r}}\sqrt{s}}{\delta}\right).$$

The proof of this lemma is provided in B.1.

Now, by utilizing this lemma, we can establish the proof of Theorem 2. However, it is important to note that the following proof is an extension of the proofs presented in Suzuki (2019); Raskutti et al. (2012) adapted to the the setting in this paper.

*Proof of Theorem 2.* Now, considering that $P_X$ is the uniform distribution, we note that $\|\cdot\|_{L^2(P_X)} = \|\cdot\|_2$. Furthermore, for any $d \ge 4$, $\delta_n > 0$, and $\epsilon_n > 0$, we define:

$$M := \mathcal{M}\left(\delta_n, \left(U\left(\mathcal{F}_{p,q}^\gamma\right)\right)^d, \|\cdot\|_2\right),$$

$$N := \mathcal{N}\left(\sqrt{\sum_{i=1}^d \sigma_i^2 \epsilon_n}, \left(U\left(\mathcal{F}_{p,q}^\gamma\right)\right)^d, \|\cdot\|_2\right).$$

Here, the covering set $\{g^1, \ldots, g^N\}$ is constructed such that for any $g \in (U(\mathcal{F}_{p,q}^\gamma))^d$, there exists $k \in [N]$ satisfying $\|g_i - g_i^k\|_2 \le \sigma_i \epsilon_n$. Moreover, let $\{f^1, \ldots, f^M\}$ be a $\delta_n$-packing of $(U(\mathcal{F}_{p,q}^\gamma))^d$.

Now, according to the proof of Theorem 2 in Raskutti et al. (2012), if we let $\Theta$ be a random uniformly distributed over the index set $[M]$ and let $X_1^n := \{x^{(i)}\}_{i=1}^n$ and $Y_1^n := \{y^{(i)}\}_{i=1}^n$, we can obtain the following lower bound:

$$
\begin{aligned}
\inf_{\hat{f}} \sup_{f^* \in (U(\mathcal{F}_{p,q}^\gamma))^\infty} \mathbb{E}_{D_n}\left[\|\hat{f} - f^*\|_{L^2(P_X)}^2\right] &\ge \inf_{\hat{f}} \sup_{f^* \in \mathcal{F}} \mathbb{E}_{D_n}\left[\|\hat{f} - f^*\|_{L^2(P_X)}^2\right] \\
&\ge \inf_{\hat{f}} \sup_{f^* \in \mathcal{F}} \frac{\delta_n^2}{2} P\left[\|\hat{f} - f^*\|_{L^2(P_X)}^2 \ge \delta_n^2/2\right] \\
&\ge \frac{\delta_n^2}{2}\left(1 - \frac{\mathbb{E}_{X_1^n}\left[I_{X_1^n}(\Theta; Y_1^n)\right] + \log 2}{\log M}\right).
\end{aligned}
$$

where, for simplicity, we define $\mathcal{F} := \left(U\left(\mathcal{F}_{p,q}^\gamma\right)\right)^d$.

Here, we evaluate the mutual information $I_{X_1^n}(\Theta; Y_1^n)$. This can be done by considering the KL-divergence, as discussed in Yang & Barron (1999); Raskutti et al. (2012):

$$I_{X_1^n}(\Theta; Y_1^n) \le \frac{1}{M}\sum_{k=1}^M \left(\log N + \frac{n}{2}\sum_{i=1}^d \frac{\|g_i^{l_i^*(k)} - f_i^k\|_n^2}{\sigma_i^2}\right) \le \log N + \frac{nd}{2}\epsilon_n^2,$$

where $l_i^*(k) \in \operatorname{argmin}_{l \in [N]} \left\| g_i^l - f_i^k \right\|_2$, and $\|f\|_n^2 := \frac{1}{n} \sum_{i=1}^n f^2(x^{(i)})$. Therefore, we can obtain the following lower bound:

$$\inf_{\hat{f}} \sup_{f^* \in (U(\mathcal{F}_{p,q}^\gamma))^\infty} \mathbb{E}_{D_n} \left[ \left\| \hat{f} - f^* \right\|_{L^2(P_X)}^2 \right] \geq \frac{\delta_n^2}{2} \left( 1 - \frac{\log N + \frac{nd}{2} \epsilon_n^2 + \log 2}{\log M} \right).$$

Thus, as shown in the proof of Theorem 4 in Suzuki (2019), by taking $\delta_n$ and $\epsilon_n$ to satisfy

$$\frac{nd}{2} \epsilon_n^2 \leq \log N, \ 8 \log N \leq \log M, \ 4 \log 2 \leq \log M,$$

the minimax rate is lower bounded by $\frac{\delta_n^2}{4}$.

To choose $\delta_n$ and $\epsilon_n$ satisfying this condition, let us first consider $\frac{nd}{2} \epsilon_n^2 \leq \log N$. For simplicity, let $\sigma = \max_{i \in [d]} \sigma_i$, and we have

$$\log \mathcal{N} \left( \sqrt{\sum_{i=1}^d \sigma_i^2} \epsilon_n, \mathcal{F}, \|\cdot\|_2 \right) \geq \log \mathcal{N} \left( \sqrt{d} \sigma \epsilon_n, \mathcal{F}, \|\cdot\|_2 \right).$$

Now, since $p \geq 2$ by assumption, according to Lemma 10, we have

$$\log \mathcal{N} \left( \sqrt{d} \sigma \epsilon_n, \mathcal{F}, \|\cdot\|_2 \right) \gtrsim s \log \frac{d}{s} + s d^{\frac{1}{ra^\dagger}} \log \left( \frac{d^{-\frac{1}{r}} \sqrt{s}}{\sqrt{d} \epsilon} \right).$$

Thus, we obtain

$$nd\epsilon_n^2 \sim s \log \frac{d}{s} + s d^{\frac{1}{ra^\dagger}} \log \left( \frac{d^{-\frac{1}{r}} \sqrt{s}}{\sqrt{d} \epsilon} \right)$$

$$\iff \epsilon_n^2 \sim \frac{1}{n} \left( \frac{s}{d} \log \frac{d}{s} + \frac{s}{d} d^{\frac{1}{ra^\dagger}} \log \left( \frac{d^{-\frac{1}{2} - \frac{1}{r}} \sqrt{s}}{\epsilon} \right) \right).$$

We want to choose $\epsilon_n^2$ that satisfies this equation. Let's consider $s$ first. A larger $s$ gives a better lower bound, so we set $s \sim d$. In this case, we have

$$\epsilon_n^2 \sim \frac{1}{n} \left( d^{\frac{1}{ra^\dagger}} \log \left( \frac{d^{-\frac{1}{r}}}{\epsilon} \right) \right).$$

Therefore, $\epsilon_n \sim n^{-\frac{a^\dagger}{2a^\dagger + 1}}$, $d \sim \epsilon_n^{-r} \sim n^{\frac{ra^\dagger}{2a^\dagger + 1}}$ would be the optimal choice.

Finally, we need to determine the relationship between $\epsilon_n$ and $\delta_n$ to satisfy $8 \log N \leq \log M$, which can be achieved by setting $\sqrt{\sum_{i=1}^d \sigma_i^2} \epsilon_n \sim \delta_n$. Thus, we obtain $\delta_n \sim n^{-\frac{(1 - \frac{r}{2}) a^\dagger}{2a^\dagger + 1}}$.

In conclusion, the lower bound is given as follows:

$$\inf_{\hat{f}} \sup_{f^* \in (U(\mathcal{F}_{p,q}^\gamma))^\infty} \mathbb{E}_{D_n} \left[ \left\| \hat{f} - f^* \right\|_{L^2(P_X)}^2 \right] \gtrsim n^{-\frac{(2-r) a^\dagger}{2a^\dagger + 1}}.$$

∎

### B.1 Proof of Lemma 10

This theorem is an extension of Lemma 4 (a) from Raskutti et al. (2012) to our settings. In order to prove this lemma, we make use of the following lemma.

**Lemma 11.** In accordance with the definition used in Theorem 1, we define and utilize $a^\dagger$. For any $i \in [d]$ and $p \geq 2$, we have

$$\log \mathcal{N} \left( \frac{\delta}{\sqrt{s_i}}, \left( U \left( \mathcal{F}_{p,q}^\gamma \right) \right)_i, \|\cdot\|_2 \right) \gtrsim d^{\frac{1}{ra^\dagger}} \log \left( \frac{d^{-\frac{1}{r}} \sqrt{s_i}}{\delta} \right),$$

where $\left( U \left( \mathcal{F}_{p,q}^\gamma \right) \right)_i$ denotes the space $U \left( \mathcal{F}_{p,q}^\gamma \right)$ with the constraint $\|f_i\|_2 \leq B_2 i^{-\frac{1}{r}}$.

The proof of this lemma is provided in B.2.

Lemma 11 provides an evaluation of the covering number for a constrained $\gamma$-smooth space. On the other hand, Lemma 10 provides an evaluation for the product space formed by combining multiple such spaces. The proof of Lemma 10 is more general, allowing for evaluations in various settings. Therefore, it is reasonable to expect that by providing the corresponding evaluations appearing in Lemma 11 in other settings, similar evaluations can be obtained for product spaces.

*Proof of Lemma 10.* First, we define

$$N_i = \mathcal{M}\left(\frac{\delta}{\sqrt{s_i}}, \left(\mathcal{F}_{p,q}^\gamma\right)_i, \|\cdot\|_2\right) - 1.$$

Next, we introduce $I_i = \{0, 1, 2, \ldots, N_i\}$ and define

$$\mathfrak{S} = \left\{ u \in \prod_{i=j}^d I_i \ \middle| \ \|u\|_0 = s \right\},$$

where we set $s = \max_{i \in [d]} s_i$.

Now, from Lemma 11, we can obtain

$$N_i = \mathcal{M}\left(\frac{\delta}{\sqrt{s_i}}, \left(\mathcal{F}_{p,q}^\gamma\right)_i, \|\cdot\|_2\right) - 1 \gtrsim d^{\frac{1}{ra\dagger}} \log\left(\frac{d^{-\frac{1}{r}}\sqrt{s_i}}{\delta}\right) - 1.$$

Therefore, $s_i = d^{\frac{2}{r}}\bar{s}$ implies that we can set

$$N_i \gtrsim d^{\frac{1}{ra\dagger}} \log\left(\frac{\sqrt{\bar{s}}}{\delta}\right) - 1 =: N$$

which results in a value independent of $i$. Note that $s = \max_i s_i = d^{\frac{2}{r}}\bar{s}$ holds.

Now, we have $|\mathfrak{S}| \gtrsim \binom{d}{s} N^d$.

For $i = [d]$, we select $\{0, f_i^1, f_i^2, \ldots, f_i^N\}$ as a $\frac{\delta}{\sqrt{s_i}}$-packing of $\left(\mathcal{F}_{p,q}^\gamma\right)_i$, and for any $u \in \mathfrak{S}$, we define

$$g^u := (g_1^{u_1}, g_2^{u_2}, \ldots, g_d^{u_d}) \in \left(\mathcal{F}_{p,q}^\gamma\right)^d,$$

where if $u_i \neq 0$, then $g_i^{u_i} = f_i^{u_i}$, and if $u_i = 0$, then $g_i^{u_i} = 0$.

Next, let's consider $g^u$ and $h^v$ that belong to $\{g^u, u \in \mathfrak{S}\}$. From the definition, we have

$$\|g^u - h^v\|_2^2 = \sum_{i=1}^d \|f_i^{u_i} - f_i^{v_i}\|_2^2 \geq \sum_{i=1}^d \frac{\delta^2}{s_i} I[u_i \neq v_i] \geq \frac{\delta^2}{s} \sum_{i=1}^d I[u_i \neq v_i].$$

Moving forward, we can follow the same reasoning as in the proof of Lemma 4 (a) in Raskutti et al. (2012), and obtain the following bound for $s \leq \frac{d}{4}$:

$$\log \mathcal{N}\left(\delta, \left(\mathcal{F}_{p,q}^\gamma\right)^d, \|\cdot\|_2\right) \gtrsim s \log \frac{d}{s} + s d^{\frac{1}{ra\dagger}} \log\left(\frac{d^{-\frac{1}{r}}\sqrt{s}}{\delta}\right).$$

Here, we substitute $\bar{s} = d^{-\frac{2}{r}}s$ into $N$ to obtain this result. ∎

## B.2 PROOF OF LEMMA 11

*Proof of Lemma 11.* First, we choose the value $s$ to satisfy $2^{\gamma(s)} = (B_2)^{-1} d^{\frac{1}{r}}$. Then, for this value of $s$, we consider the corresponding ball:

$$\left\{ \delta_s(f) \mid \|\delta_s(f)\|_{\mathcal{F}_{p,q}^\gamma} \leq 1 \right\}.$$

This ball satisfies the decay assumption $\|f_i\|_2 \leq B_2 i^{-\frac{1}{r}}$. In fact,

$$\|\delta_s(f)\|_{\mathcal{F}_{p,q}^{\gamma}} = \left( \sum_s \left( 2^{\gamma(s)} \|\delta_s(\delta_s(f))\|_p \right)^q \right)^{\frac{1}{q}} = 2^{\gamma(s)} \|\delta_s(f)\|_p \leq 1$$

holds because of $f \in U\left( \mathcal{F}_{p,q}^{\gamma} \right)$. Therefore, if $p \geq 2$,

$$\|\delta_s(f)\|_2 \leq \|\delta_s(f)\|_p \leq 2^{-\gamma(s)} = B_2 d^{-\frac{1}{r}} \leq B_2 i^{-\frac{1}{r}}.$$

Because this ball is topologically equivalent to a Euclidean ball with dimension $2^s$ and radius $B_2 d^{-\frac{1}{r}}$, we have

$$\mathcal{N}\left( \frac{\delta}{\sqrt{s_i}}, \left( \mathcal{F}_{p,q}^{\gamma} \right)_i, \|\cdot\|_2 \right) \geq \mathcal{N}\left( \frac{\delta}{\sqrt{s_i}}, B_2 d^{-\frac{1}{r}} \mathbb{B}^{2^s}, \|\cdot\|_2 \right)$$

$$= \mathcal{N}\left( \frac{\delta}{B_2 d^{-\frac{1}{r}} \sqrt{s_i}}, \mathbb{B}^{2^s}, \|\cdot\|_2 \right)$$

$$\geq \left( \frac{B_2 d^{-\frac{1}{r}} \sqrt{s_i}}{\delta} \right)^{2^s}.$$

Next, let us consider a specific value for $s$. In the case of mixed smoothness, we assign values to $s$ only for the index corresponding to the minimum value of $a$, while setting the rest of the indices to $s_i = 0$ in order to maximize the right-hand side of the inequality. In this scenario, we obtain

$$2^s = \left( 2^{\gamma(s)} \right)^{\frac{1}{a^\dagger}} = (B_2)^{-\frac{1}{a^\dagger}} d^{\frac{1}{r a^\dagger}}.$$

Similarly, in the case of anisotropic smoothness, by choosing $i'$ as the index corresponding to the minimum value of $a$, we set $s_i = s_{i'} \frac{a_{i'}}{a_i}$ to maximize $2^s$. Therefore, we have

$$2^s = 2^{s_{i'} a_{i'} \sum_{i=1}^{\infty} \frac{1}{a_i}} = 2^{\frac{\gamma(s)}{a^\dagger}},$$

which coincides with the case of mixed smoothness. Thus, we can conclude that

$$\log \mathcal{N}\left( \frac{\delta}{\sqrt{s_i}}, \left( \mathcal{F}_{p,q}^{\gamma} \right)_i, \|\cdot\|_2 \right) \geq 2^s \log \left( \frac{B_2 d^{-\frac{1}{r}} \sqrt{s_i}}{\delta} \right) \gtrsim d^{\frac{1}{r a^\dagger}} \log \left( \frac{d^{-\frac{1}{r}} \sqrt{s_i}}{\delta} \right).$$

∎

## C  Proof of Theorem 4

Before delving into the discussion of this theorem, we present here the precise definition of linear estimators briefly introduced at the beginning of Section 4.

**Definition 11** (Definition of Linear Estimators). Let $D_n = \left( x^{(i)}, y^{(i)} \right)_{i=1}^{n}$ be the data set. An estimator $\hat{f}$ based on $D_n$ is said to be a linear estimator if it can be expressed in the following form:

$$\forall i \in \mathbb{N}: \quad \left( \hat{f}(x) \right)_i = \sum_{j=1}^{n} y_i^{(j)} \varphi_{i,j}(x; x^n).$$

Here, we assume $\mathbb{E}\left[ \|\varphi_{i,j}(\cdot; x^n)\|_2^2 \right] < \infty$.

The theorem is grounded in the notion that, as stated at the outset of Section 4.1, the non-convex nature of the union of $\gamma$-smooth spaces for varying values of $a$ implies that linear estimators are likely to experience sub-optimal performance. This intuition is based on Theorem 3.3 in Hayakawa & Suzuki (2020). Here, we present this theorem as an extension applicable to the setting of linear estimators in this paper.

**Theorem 12** (Extension of Theorem 3.3 in Hayakawa & Suzuki (2020)). For linear estimators (Definition 11) in a certain space $\mathcal{F}^\infty$, the estimation error has the following relationship:

$$\inf_{\hat{f}:\text{linear}} \sup_{f^\circ \in \mathcal{F}^\infty} R\left(\hat{f},\, f^\circ\right) = \inf_{\hat{f}:\text{linear}} \sup_{f^\circ \in \text{Conv}(\mathcal{F}^\infty)} R\left(\hat{f},\, f^\circ\right),$$

where $R\left(\hat{f},\, f^\circ\right) := \mathbb{E}_{P^n}\left[\left\|\hat{f} - f^\circ\right\|_{P_X}^2\right]$.

The proof is provided in C.1.

In order to utilize this theorem in the current proof, we require the result of taking the Convex Hull of the space $U\left((\mathrm{F}_{2,2}\left(\Gamma\right))^\infty\right)$. The following lemma presents the result that provides it.

**Lemma 13.** The Convex Hull of $U\left(\mathrm{F}_{2,2}\left(\Gamma\right)\right)$ is given by:

$$\text{Conv}\left(U\left(\mathrm{F}_{2,2}\left(\Gamma\right)\right)\right) = U\left(\mathcal{F}_{2,1}^{\min_{a \in \Gamma} \gamma_a}\right).$$

Here, $\gamma = \min_{a \in \Gamma} \gamma_a$ is denoted as $\gamma\left(s\right) := \min_{a \in \Gamma} \gamma_a\left(s\right)$, indicating that the minimum is taken over $a$ for each $s$.

The proof is provided in C.2.

This lemma provides a result for the space $U\left(\mathrm{F}_{2,2}\left(\Gamma\right)\right)$, but the space of interest in our discussion, $U\left((\mathrm{F}_{2,2}\left(\Gamma\right))^\infty\right)$, is simply a Cartesian product of the space $U\left(\mathrm{F}_{2,2}\left(\Gamma\right)\right)$. Therefore, we can observe that

$$\text{Conv}\left(U\left((\mathrm{F}_{2,2}\left(\Gamma\right))^\infty\right)\right) = \left(U\left(\mathcal{F}_{2,1}^{\min_{a \in \Gamma} \gamma_a}\right)\right)^\infty.$$

Furthermore, following the approach used in the proof of Theorem 2, we also seek to establish a lower bound using the techniques from Suzuki (2019); Raskutti et al. (2012). To this end, we require a lower bound on the covering number for the space under consideration, and we present the following lemma regarding that.

**Lemma 14** (Covering Number of $\mathcal{F}_{2,1}^\gamma$). From Definition 8, let $\Gamma = \Gamma\left(Q_\epsilon^m\right)$, and let $\gamma_a\left(s\right) = \langle a, s \rangle$ and $a^\star = \underline{a}$. Also, let $\Gamma = \Gamma\left(Q_\epsilon^a\right)$, $\gamma_a\left(s\right) = \max_i \{a_i s_i\}$, and $a^\star = \tilde{a}$. Here, $\gamma = \min_{a \in \Gamma} \gamma_a$. In this case, the lower bound on the covering number of $\mathcal{F}_{2,1}^\gamma$ is given by:

$$\log \mathcal{N}\left(\epsilon, U\left(\mathcal{F}_{2,1}^\gamma\right), \|\cdot\|_2\right) \gtrsim \epsilon^{-\frac{1+c}{a^\star}}.$$

The proof is provided in C.3.

By combining these lemmas and theorem, we can prove Theorem 4 as follows.

*Proof of Theorem 4.* First, the estimation error we want to evaluate can be transformed as follows:

$$\inf_{\hat{f}:\text{linear}} \sup_{f^\circ \in U((\mathrm{F}_{2,2}(\Gamma))^\infty)} \mathbb{E}_{D_n}\left[\left\|\hat{f} - f^\circ\right\|_{L^2(P_X)}\right]$$

$$= \inf_{\hat{f}:\text{linear}} \sup_{f^\circ \in \text{Conv}(U((\mathrm{F}_{2,2}(\Gamma))^\infty))} \mathbb{E}_{D_n}\left[\left\|\hat{f} - f^\circ\right\|_{L^2(P_X)}\right]$$

$$= \inf_{\hat{f}:\text{linear}} \sup_{f^\circ \in \left(U\left(\mathcal{F}_{2,1}^\gamma\right)\right)^\infty} \mathbb{E}_{D_n}\left[\left\|\hat{f} - f^\circ\right\|_{L^2(P_X)}\right]$$

$$\gtrsim \inf_{\hat{f}} \sup_{f^\circ \in U\left(\mathcal{F}_{2,1}^\gamma\right)} \mathbb{E}_{D_n}\left[\left\|\hat{f} - f^\circ\right\|_{L^2(P_X)}\right],$$

where in the second line we used Theorem 12, and in the third line we used Lemma 13.

In other words, we ultimately need to evaluate $\inf_{\hat{f}} \sup_{f^\circ \in U\left(\mathcal{F}_{2,1}^\gamma\right)} \mathbb{E}_{D_n}\left[\left\|\hat{f} - f^\circ\right\|_{L^2(P_X)}\right]$. To evaluate this, we first let

$$M := \mathcal{M}\left(\delta_n, U\left(\mathcal{F}_{2,1}^\gamma\right), \|\cdot\|_2\right), \quad N := \mathcal{N}\left(\epsilon_n, U\left(\mathcal{F}_{2,1}^\gamma\right), \|\cdot\|_2\right).$$

Then, according to Suzuki (2019), we have:

$$\inf_{\hat{f}} \sup_{f^\circ \in U\left(\mathcal{F}_{2,1}^\gamma\right)} \mathbb{E}_{D_n}\left[\left\|\hat{f} - f^\circ\right\|_{L^2(P_X)}\right] \gtrsim \frac{\delta_n^2}{2}\left(1 - \frac{\log N + \frac{n}{2\sigma_1^2}\epsilon_n^2 + \log 2}{\log M}\right).$$

If we select $\delta_n$, $\epsilon_n$ satisfying

$$\frac{n}{2\sigma_1^2}\epsilon_n^2 \leq \log N, \ 8\log N \leq \log M, \ 4\log 2 \leq \log M,$$

the minimax rate is lower bounded by $\frac{\delta_n^2}{4}$.

Here, from Lemma 14, we have:

$$\log \mathcal{N}\left(\epsilon, U\left(\mathcal{F}_{2,1}^\gamma\right), \|\cdot\|_2\right) \gtrsim \epsilon^{-\frac{1+c}{a^\star}}.$$

Therefore, by choosing $\epsilon_n$ and $\delta_n$ such that:

$$\delta_n \sim \epsilon_n \sim n^{-\frac{a^\star}{2a^\star + 1 + c}},$$

we ultimately obtain:

$$\inf_{\hat{f}} \sup_{f^\circ \in U\left(\mathcal{F}_{2,1}^\gamma\right)} \mathbb{E}_{D_n}\left[\left\|\hat{f} - f^\circ\right\|_{L^2(P_X)}\right] \gtrsim \delta_n^2 \gtrsim n^{-\frac{2a^\star}{2a^\star + 1 + c}}.$$

∎

## C.1 Proof of Theorem 12

*Proof of Theorem 12.* Now, we define

$$R\left(\hat{f}, f^\circ\right) := \mathbb{E}_{P^n}\left[\left\|\hat{f} - f^\circ\right\|_{P_X}^2\right], \ \tilde{R}\left(\hat{f}_i, f_i^\circ\right) := \mathbb{E}_{P^n}\left[\left\|\hat{f}_i - f_i^\circ\right\|_{P_X}^2\right].$$

In this case, we obtain:

$$R\left(\hat{f}, f^\circ\right) = \mathbb{E}_{P^n}\left[\left\|\hat{f} - f^\circ\right\|_{P_X}^2\right] = \mathbb{E}_{P^n}\left[\mathbb{E}_{P_X}\left[\left\|\hat{f}(X) - f^\circ(X)\right\|_{\ell^2}^2\right]\right]$$

$$= \mathbb{E}_{P^n}\left[\mathbb{E}_{P_X}\left[\sum_{i=1}^\infty \left(\hat{f}_i(X) - f_i^\circ(X)\right)^2\right]\right] = \sum_{i=1}^\infty \mathbb{E}_{P^n}\left[\mathbb{E}_{P_X}\left[\left(\hat{f}_i(X) - f_i^\circ(X)\right)^2\right]\right]$$

$$= \sum_{i=1}^\infty \tilde{R}\left(\hat{f}_i, f_i^\circ\right).$$

Here, $\hat{f}_i$ represents a one-dimensional output linear estimator, defined in Hayakawa & Suzuki (2020). Therefore, as proven in the proof of Theorem 3.3 in Hayakawa & Suzuki (2020), $\tilde{R}\left(\hat{f}_i, \cdot\right)$ is a convex function. In other words, for $f^\circ$, $g^\circ \in \mathcal{F}^\infty$ and $h^\circ := tf^\circ + (1-t)g^\circ$, we have:

$$\tilde{R}\left(\hat{f}_i, h_i^\circ\right) \leq t\tilde{R}\left(\hat{f}_i, f_i^\circ\right) + (1-t)\tilde{R}\left(\hat{f}_i, g_i^\circ\right).$$

Hence,

$$R\left(\hat{f}, h^\circ\right) = \sum_{i=1}^\infty \tilde{R}\left(\hat{f}_i, h_i^\circ\right) \leq t\sum_{i=1}^\infty \tilde{R}\left(\hat{f}_i, f_i^\circ\right) + (1-t)\sum_{i=1}^\infty \tilde{R}\left(\hat{f}_i, g_i^\circ\right)$$

$$= tR\left(\hat{f}, f^\circ\right) + (1-t)R\left(\hat{f}, g^\circ\right)$$

holds. Therefore, $R\left(\hat{f}, \cdot\right)$ is a convex function. Therefore,

$$\inf_{\hat{f}:\text{linear}} \sup_{f^\circ \in \mathcal{F}^\infty} R\left(\hat{f}, f^\circ\right) \geq \inf_{\hat{f}:\text{linear}} \sup_{f^\circ \in \text{Conv}(\mathcal{F}^\infty)} R\left(\hat{f}, f^\circ\right)$$

can be asserted. And since the converse is evident, we can conclude the statement of this theorem.

∎

## C.2 PROOF OF LEMMA 13

In order to prove this lemma, we consider an alternative representation of the $\gamma$-smooth space using a method similar to the one employed in describing Besov spaces based on wavelet expansions, as demonstrated in Donoho & Johnstone (1998) and Meyer (1993). Here, we adapt a similar approach to redefine the $\gamma$-smooth space.

**Theorem 15** (Alternative representation of $\gamma$-smooth space). For the norm $\|\cdot\|_{\mathcal{F}_{p,q}^{\gamma}}$ of the $\gamma$-smooth space $\mathcal{F}_{p,q}^{\gamma}$, the following relationship holds:

$$\|f\|_{\mathcal{F}_{p,q}^{\gamma}} \asymp \left( \sum_{s \in \mathbb{N}_0^{\infty}} \left( 2^{\gamma(s) - \left(\frac{s}{p} - \frac{s}{2}\right)} \left( \sum_{l \in J(s)} |\theta_l|^p \right)^{\frac{1}{p}} \right)^q \right)^{\frac{1}{q}},$$

where $J(s) = \left\{ l \in \mathbb{Z}_0^{\infty} : \lfloor 2^{s_i - 1} \rfloor \leq |l_i| < 2^{s_i} \right\}$ and $\theta_l = \langle f, \psi_l \rangle$.

In other words, by considering the following space, we obtain an alternative representation of the $\gamma$-smooth space $\mathcal{F}_{p,q}^{\gamma}$:

$$\Theta_{p,q}^{\gamma}(C) := \left\{ \theta : \|\theta\|_{\mathbf{f}_{p,q}^{\gamma}} \leq C \right\},$$

$$\|\theta\|_{\mathbf{f}_{p,q}^{\gamma}} := \left( \sum_{s \in \mathbb{N}_0^{\infty}} \left( 2^{\gamma(s) - \left(\frac{s}{p} - \frac{s}{2}\right)} \left( \sum_{l \in J(s)} |\theta_l|^p \right)^{\frac{1}{p}} \right)^q \right)^{\frac{1}{q}}.$$

The proof is provided in C.2.1.

Hereafter, based on this theorem, we treat $U\left(\mathcal{F}_{p,q}^{\gamma}\right)$ as $\Theta_{p,q}^{\gamma} := \Theta_{p,q}^{\gamma}(1)$.

*Proof of Lemma 13.* First, the parameter representation of $U\left(\mathrm{F}_{2,2}(\Gamma)\right)$ is described as follows:

$$\bigcup_{a \in \Gamma} \Theta_{2,2}^{\gamma_a} = \bigcup_{a \in \Gamma} \left\{ \theta : \left( \sum_{s \in \mathbb{N}_0^{\infty}} \left( 2^{\gamma_a(s)} \left( \sum_{l \in J(s)} |\theta_l|^2 \right)^{\frac{1}{2}} \right)^2 \right)^{\frac{1}{2}} \leq 1 \right\}.$$

Now, let us define $\theta_s := (\theta_l)_{l \in J(s)}$. Next, let us consider fixing a particular $s$ and setting $\theta_{s'} = (0)_{l \in J(s')}$ for all $s' \neq s$. Then, we have

$$2^{\gamma_a(s)} \|\theta_s\|_2 \leq 1.$$

On the other hand, since we are considering $\bigcup_{a \in \Gamma}$, as the set of endpoints, we can consider:

$$2^{\min_{a \in \Gamma} \gamma_a(s)} \|\theta_s\|_2 \leq 1.$$

Now, let us vary $s$ and consider taking the convex hull over all $\theta_s$ that satisfy the above condition. Specifically, we consider a convex combination $\theta = \sum_{s \in \mathbb{N}_0^{\infty}} \delta_s \theta_s$ using $\delta_s \in \mathbb{R}$ satisfying $\sum_{s \in \mathbb{N}_0^{\infty}} \delta_s \leq 1$ and $\delta_s > 0$. For this $\theta$, we consider the $\|\cdot\|_{\mathbf{f}_{2,1}^{\min_{a \in \Gamma} \gamma_a}}$ norm, and we have:

$$\|\theta\|_{\mathbf{f}_{2,1}^{\min_{a \in \Gamma} \gamma_a}} = \sum_{s \in \mathbb{N}_0^{\infty}} 2^{\min_{a \in \Gamma} \gamma_a(s)} \|\delta_s \theta_s\|_2$$

$$= \sum_{s \in \mathbb{N}_0^{\infty}} \delta_s 2^{\min_{a \in \Gamma} \gamma_a(s)} \|\theta_s\|_2$$

$$\leq \sum_{s \in \mathbb{N}_0^{\infty}} \delta_s \leq 1.$$

Thus, we obtain the relationship stated in the theorem. ∎

### C.2.1 PROOF OF THEOREM 15

*Proof of Lemma 15.* The proof utilizes the technique presented in Meyer (1993). Now, the norm of $\mathcal{F}_{p,q}^{\gamma}$ is

$$\|f\|_{\mathcal{F}_{p,q}^{\gamma}} = \left( \sum_{s \in \mathbb{N}_0^{\infty}} \left( 2^{\gamma(s)} \|\delta_s(f)\|_p \right)^q \right)^{\frac{1}{q}}.$$

Therefore, let us proceed to evaluate $\|\delta_s(f)\|_p$. Let us define $\theta_l = \langle f, \psi_l \rangle$. Then, we can express $\delta_s(f)$ as a

$$\delta_s(f) = \sum_{l \in J(s)} \langle f, \psi_l \rangle \psi_l = \sum_{l \in J(s)} \theta_l \psi_l.$$

Here, since the following inequality holds:

$$\theta_l = \langle \delta_s(f), \psi_l \rangle = \int_{\mathcal{M}} \delta_s(f)(x) \psi_l(x) \, d\lambda^{\infty}(x),$$

if we set $\frac{1}{p} + \frac{1}{q} = 1$, then

$$|\theta_l| \leq \int_{\mathcal{M}} |\delta_s(f)(x)| |\psi_l(x)|^{\frac{1}{p}} |\psi_l(x)|^{\frac{1}{q}} \, d\lambda^{\infty}(x)$$

$$\leq \left( \int_{\mathcal{M}} |\delta_s(f)(x)|^p |\psi_l(x)| \, d\lambda^{\infty}(x) \right)^{\frac{1}{p}} \left( \int_{\mathcal{M}} |\psi_l(x)| \, d\lambda^{\infty}(x) \right)^{\frac{1}{q}}$$

is obtained. Now, by the definition of $\psi_l(x)$, we obtain following evaluation:

$$|\psi_l(x)| = \left| \prod_{i=1}^{\infty} \psi_{l_i}(x_i) \right| \leq \sqrt{2}^{\|s\|_0} = 2^{\frac{\|s\|_0}{2}}.$$

Therefore, $|\theta_l|$ is evaluated as follows:

$$|\theta_l| \leq 2^{\frac{\|s\|_0}{2}} \|\delta_s(f)\|_p,$$

$$\therefore \left( \sum_{l \in J(s)} |\theta_l|^p \right)^{\frac{1}{p}} \leq \left( 2^{\frac{p\|s\|_0}{2}} \|\delta_s(f)\|_p^p |J(s)| \right)^{\frac{1}{p}}$$

$$= 2^{\frac{\|s\|_0}{2}} |J(s)|^{\frac{1}{p}} \|\delta_s(f)\|_p.$$

Furthermore, regarding $|J(s)|$, considering all possible choices for each $l_i$, which amounts to $2^{s_i}$ possibilities, we can conclude that $|J(s)| = 2^s$. Combining the above observations, we can ultimately evaluate it as follows:

$$\left( \sum_{l \in J(s)} |\theta_l|^p \right)^{\frac{1}{p}} \leq 2^{\frac{\|s\|_0}{2} + \frac{s}{p}} \|\delta_s(f)\|_p. \tag{6}$$

We now proceed to establish the reverse inequality in a similar manner. By employing a similar evaluation as before, we can derive the following inequality:

$$|\delta_s(f)(x)| \leq \sum_{l \in J(s)} |\theta_l| |\psi_l(x)| \leq \left( \sum_{l \in J(s)} |\theta_l|^p |\psi_l(x)| \right)^{\frac{1}{p}} \left( \sum_{l \in J(s)} |\psi_l(x)| \right)^{\frac{1}{q}}$$

$$\leq 2^{\frac{\|s\|_0}{2p}} \left( \sum_{l \in J(s)} |\theta_l|^p \right)^{\frac{1}{p}} \left( 2^{\frac{\|s\|_0}{2}+s} \right)^{\frac{1}{q}} \leq 2^{\frac{\|s\|_0}{2p} + \frac{1}{q}\left(\frac{\|s\|_0}{2}+s\right)} \left( \sum_{l \in J(s)} |\theta_l|^p \right)^{\frac{1}{p}}$$

$$\leq 2^{\frac{\|s\|_0}{2p} + \left(1 - \frac{1}{p}\right)\left(\frac{\|s\|_0}{2}+s\right)} \left( \sum_{l \in J(s)} |\theta_l|^p \right)^{\frac{1}{p}} \leq 2^{\frac{\|s\|_0}{2} - \frac{s}{p}+s} \left( \sum_{l \in J(s)} |\theta_l|^p \right)^{\frac{1}{p}}.$$

Then, $\|\delta_s(f)\|_p$ is evaluated as follows:

$$\|\delta_s(f)\|_p = \left(\int_{\mathcal{M}} |\delta_s(f)(x)|^p\right)^{\frac{1}{p}}$$

$$\leq 2^{\frac{\|s\|_0}{2} - \frac{s}{p} + s} \left(\sum_{l \in J(s)} |\theta_l|^p\right)^{\frac{1}{p}}. \tag{7}$$

By combining equations Eq (6) and Eq (7), we obtain the following relationship:

$$2^{-\frac{\|s\|_0}{2} - \frac{s}{p}} \left(\sum_{l \in J(s)} |\theta_l|^p\right)^{\frac{1}{p}} \leq \|\delta_s(f)\|_p \leq 2^{\frac{\|s\|_0}{2} - \frac{s}{p} + s} \left(\sum_{l \in J(s)} |\theta_l|^p\right)^{\frac{1}{p}}$$

$$\therefore \|\delta_s(f)\|_p \asymp 2^{-\frac{s}{p} + \frac{s}{2}} \left(\sum_{l \in J(s)} |\theta_l|^p\right)^{\frac{1}{p}}.$$

Substituting this into the initial expression for $\|f\|_{\mathcal{F}_{p,q}^\gamma}$ allows us to establish the desired relationship. ∎

## C.3 PROOF OF LEMMA 14

*Proof of Lemma 14.* Let us consider the case where we define $\Gamma = \Gamma(Q_\epsilon^m)$, $\gamma_a(s) = \langle a, s \rangle$, and $a^\star = \underline{a}$. For simplicity, we denote $Q_\epsilon := Q_\epsilon^m$.

Now, we define the set $I(Q_\epsilon)$ as follows:

$$I(Q_\epsilon) := \left\{ s \in \mathbb{N}_0^\infty \,\middle|\, s_i = \begin{cases} \lfloor \frac{T}{\underline{a}} \rfloor & \left(i \in \left[2^{\frac{a}{Q_\epsilon}}\right]\right) \\ 0 & \left(i \notin \left[2^{\frac{a}{Q_\epsilon}}\right]\right) \end{cases} \right\}.$$

We now evaluate the restricted subspace corresponding to the coordinates associated with $s$ belonging to this set. In other words, for $s \notin I(Q_\epsilon)$, we set the coordinates $\theta_{s,l}$ corresponding to $s$ and $l$ to be $0$. The dimension of this subspace, denoted as $d$, is given by:

$$d = \sum_{s \in I(Q_\epsilon)} \sum_{l \in J(s)} = \sum_{s \in I(Q_\epsilon)} 2^s.$$

Then, the restricted subspace of $U\left(\mathcal{F}_{2,1}^\gamma\right)$ can be expressed as:

$$\left\{ \theta : \sum_{s \in I(Q_\epsilon)} 2^{\gamma(s)} \left(\sum_{l \in J(s)} |\theta_l|^2\right)^{\frac{1}{2}} \leq 1 \right\}.$$

On the other hand, by using Cauchy–Schwarz inequality, we obtain

$$\sum_{s \in I(Q_\epsilon)} 2^{\gamma(s)} \left(\sum_{l \in J(s)} |\theta_l|^2\right)^{\frac{1}{2}} \leq \sqrt{\sum_{s \in I(Q_\epsilon)} 1 \left(\sum_{s \in I(Q_\epsilon)} \left(2^{\gamma(s)} \left(\sum_{l \in J(s)} |\theta_l|^2\right)^{\frac{1}{2}}\right)^2\right)^{\frac{1}{2}}}.$$

Then, if we define $d' := \sum_{s \in I(Q_\epsilon)} 1$, the space

$$\left\{ \theta : \sum_{s \in I(Q_\epsilon)} 2^{2\gamma(s)} d' \sum_{l \in J(s)} |\theta_l|^2 \leq 1 \right\}.$$

also be a subspace of the restricted subspace of $U\left(\mathcal{F}_{2,1}^\gamma\right)$.

Next, we consider the covering number of this space. To achieve this, we want to determine the value of $T$ such that a ball with radius $\epsilon$ can fit inside this space. Based on the assumption that $2^{-2\gamma(s)} > 2^{-2T}$, we have:

$$\frac{2^{-2T}}{d'} \geq \epsilon^2.$$

We need to choose the largest possible value of $T$ that satisfies this inequality.

Here, $d'$ can be derived from the definition of $I(Q_\epsilon)$ as $d' = \lfloor 2^{\frac{a}{Q_\epsilon}} \rfloor \leq 2^{\frac{a}{Q_\epsilon}}$. Therefore, we obtain:

$$\frac{2^{-2T}}{d'} \geq 2^{-2T - \frac{a}{Q_\epsilon}}.$$

By setting $T = \frac{2a-1-c}{2a-1} \log_2 \epsilon^{-1}$ and substituting this and the definition of $Q_\epsilon$, we have:

$$\frac{2^{-2T}}{d'} \geq 2^{-2T - \frac{a}{Q_\epsilon}} = 2^{-\left(2\frac{2a-1-c}{2a-1} + \frac{2c}{2a-1}\right) \log_2 \epsilon^{-1}} = \epsilon^2.$$

This shows that by setting $T$ in this way, a ball with radius $\epsilon$ is guaranteed to be contained within the subspace.

The remaining task is to evaluate the covering number of the ball with radius $\epsilon$. According to (5.9) in Wainwright (2019), we have:

$$\log \mathcal{N}\left(\frac{\epsilon}{3}, \mathbb{B}^d(\epsilon), \|\cdot\|_2\right) \geq d \log 3.$$

Here, $d$ can be expressed as follows based on the definition:

$$d = \sum_{s \in I(Q_\epsilon)} 2^s \simeq \sum_{s \in I(Q_\epsilon)} 2^{\frac{T}{a}} \simeq 2^{\frac{a}{Q_\epsilon} + \frac{T}{a}}.$$

By substituting the values of $T$ and $Q_\epsilon$, we obtain:

$$d \simeq 2^{\frac{a}{Q_\epsilon} + \frac{T}{a}} = 2^{\left(\frac{2c}{2a-1} + \frac{2a-1-c}{a(2a-1)}\right) \log_2 \epsilon^{-1}} = \epsilon^{-\frac{1+c}{a}}.$$

Thus, we obtain the desired result.

In addition, let us consider the case where $\Gamma = \Gamma(Q_\epsilon^a)$, $\gamma_a(s) = \max_i \{a_i s_i\}$, and $a^\star = \tilde{a}$. Now, in order to analyze $\min_{a \in \Gamma(Q_\epsilon)}$, it suffices to consider the case when $a_{i_j} = \underline{a} j^\eta$, because the following condition

$$\|a\|_{wl^\eta} \leq \underline{a}^{-1} \iff a_{i_j} \geq \underline{a} j^\eta \quad (\forall j = 1, 2, \dots)$$

holds. Now, let us consider the following setting as a specific example of $s$ satisfying $\gamma(s) < T$:

$$s = \left(\left\lfloor \frac{T}{\underline{a}} \right\rfloor, \left\lfloor \frac{T}{\underline{a} 2^\eta} \right\rfloor, \left\lfloor \frac{T}{\underline{a} 3^\eta} \right\rfloor, \dots, \left\lfloor \frac{T}{\underline{a} K^\eta} \right\rfloor, 0, \dots\right),$$

where $K = \left\lfloor 2^{\frac{a}{Q_\epsilon}} \right\rfloor$.

In this case, for each $s$, the $a$ that achieves the minimum of $\gamma(s) = \min_{a \in \Gamma(Q_\epsilon)} \gamma_a(s)$ corresponds to a mapping between the elements of $s$ in descending order and the elements of $a$ in ascending order. Therefore, we have $\gamma(s) = \max_j a_{i_j} s_j$. Furthermore, considering that

$$\underline{a} i^\eta \left\lfloor \frac{T}{\underline{a} i^\eta} \right\rfloor < T \quad (\forall i = 1, 2, \dots),$$

the defined $s$ satisfies the condition $\gamma(s) < T$. Moreover, such $s$ can be rearranged by changing the indices. If we denote the index corresponding to the minimum value of $s$ as $i_1$, it is sufficient for $i_1$ to be within the range where $\underline{a} \geq Q_\epsilon \log_2 i_1$ holds. Therefore, there are at least $2^{\frac{a}{Q_\epsilon}}$ possible rearrangements. Let $I(Q_\epsilon)$ represent the set of $2^{\frac{a}{Q_\epsilon}}$ rearranged $s$.

Under this setting, following the previous discussion, we want to choose the largest $T$ that satisfies:

$$\frac{2^{-2T}}{d'} \geq \epsilon^2.$$

Now, since $d' \leq 2^{\frac{a}{Q_\epsilon}}$ holds based on its definition, we have $\frac{2^{-2T}}{d'} \geq 2^{-2T-\frac{a}{Q_\epsilon}}$. Therefore, substituting $T = \left(1 - \frac{2(1+c)}{\tilde{a}}\right) \log_2 \epsilon^{-1}$ along with the definition of $Q_\epsilon$, we obtain:

$$\frac{2^{-2T}}{d'} \geq 2^{-2T-\frac{a}{Q_\epsilon}}$$
$$= 2^{-\frac{1}{2\eta\tilde{a}}\left(\left(4\eta\tilde{a}-6\eta(1+c)+\frac{2(1+c)}{\tilde{a}}-1\right)\log_2 \epsilon^{-1}+\underline{a}^{1-1/\eta}\left(\log_2 \epsilon^{-1}\right)^{1/\eta}\right)}.$$

Now, considering the definition of $c$, we have $\frac{2(1+c)}{\tilde{a}} - 1 < 0$. Additionally, when $\epsilon$ is sufficiently small, the value of $\log_2 \epsilon^{-1}$ can be considered sufficiently large compared to $\left(\log_2 \epsilon^{-1}\right)^{1/\eta}$. Therefore, we have $\left(-6\eta\left(1+c\right)+\frac{2(1+c)}{\tilde{a}}-1\right)\log_2 \epsilon^{-1} + \underline{a}^{1-1/\eta}\left(\log_2 \epsilon^{-1}\right)^{1/\eta} < 0$. Therefore, we obtain:

$$\frac{2^{-2T}}{d'} \geq 2^{-\frac{1}{2\eta\tilde{a}}\left(\left(4\eta\tilde{a}-6\eta(1+c)+\frac{2(1+c)}{\tilde{a}}-1\right)\log_2 \epsilon^{-1}+\underline{a}^{1-1/\eta}\left(\log_2 \epsilon^{-1}\right)^{1/\eta}\right)} \geq 2^{-2\log_2 \epsilon^{-1}} = \epsilon^2.$$

Furthermore, as for the evaluation of $d$, we have:

$$d = \sum_{\gamma(s)<T} 2^s \geq 2^{\frac{a}{Q_\epsilon}} 2^{\sum_{1\leq i\leq (T/\underline{a})^{1/\eta}}\lfloor\frac{T}{\underline{a}i^\eta}\rfloor} \geq 2^{\frac{a}{Q_\epsilon}} 2^{\sum_{1\leq i\leq (T/\underline{a})^{1/\eta}}\frac{T}{2\underline{a}i^\eta}}$$

$$\geq 2^{\frac{a}{Q_\epsilon}+\frac{T}{2\tilde{a}}\frac{1}{\eta}\left(1-(T/\underline{a})^{\frac{1-\eta}{\eta}}\right)}.$$

Then, we obtain

$$d \gtrsim 2^{\frac{a}{Q_\epsilon}+\frac{T}{2\tilde{a}}\frac{1}{\eta}\left(1-(T/\underline{a})^{\frac{1-\eta}{\eta}}\right)}$$
$$= 2^{\frac{1}{2\eta\tilde{a}}\left(\left(\frac{2(1+c)(\eta\tilde{a}+1)+\tilde{a}-2(1+c)}{\tilde{a}}-1\right)\log_2 \epsilon^{-1}+\underline{a}^{1-1/\eta}\left(1-\left(1-\frac{2(1+c)}{\tilde{a}}\right)^{1/\eta}\right)\left(\log_2 \epsilon^{-1}\right)^{1/\eta}\right)}$$
$$> \epsilon^{-\frac{1+c}{\tilde{a}}},$$

where we have utilized the fact that $0 < \frac{2(1+c)}{\tilde{a}} < 1$, which follows from the definition of $c$. Consequently, we obtain the desired result in this case as well. ∎

# D    PROOFS OF COROLLARY 6 AND THEOREM 7

## D.1    PROOF OF COROLLARY 6

*Proof of Corollary 6.* Let us consider the case where we define $\Gamma = \Gamma(Q_\epsilon^m)$, $\gamma_a(s) = \langle a, s \rangle$, and $a^\star = \underline{a}$. For simplicity, we denote $Q_\epsilon := Q_\epsilon^m$.

In this study, we consider $Q_\epsilon$ as a value that varies and is taken to be $Q_\epsilon \sim -1/\log \epsilon$. Because we compare the results with those of linear estimators, following the definition in Theorem 4, we set $\epsilon \sim n^{-\frac{a^\star}{2a^\star+1+c}}$. Consequently, we have $1/Q_\epsilon \sim \log n$ in this case.

Now, the rates for $\left(U(\mathcal{F}_{2,2}^{\gamma_a})\right)^\infty$ are provided in Theorem 3. By examining its proof, we can observe that it utilizes Theorem 10 and 14 from Okumoto & Suzuki (2021), which are results concerning one-dimensional outputs. However, in the proof presented in Okumoto & Suzuki (2021), $Q_\epsilon$ is treated as a constant and ignored. Therefore, it is necessary to consider the impact of $Q_\epsilon$ not being a constant in this case.

Specifically, $Q_\epsilon$ is used in the definition of the number of layers $L' = \lceil \frac{T}{Q_\epsilon} \rceil$. Here, $T$ takes the value $T \sim \log n$, meaning that in our setup, we have $L' \sim (\log n)^2$, which differs from the value of $L' \sim \log n$ mentioned in Theorem 14 of Okumoto & Suzuki (2021).

Therefore, we need to consider the influence of the change in $L'$ on the rate of estimation error. In Okumoto & Suzuki (2021), the authors obtained an upper bound on the covering number of the class to which dilated CNNs belong by substituting the values of each neural network's parameters into the expression:

$$(S + W'C)(L + L') \log \left(\frac{LL'(B' \vee 1)(B \vee 1)CW'W}{\delta}\right).$$

By using $L = \max\left\{T^{\frac{2}{\eta}}, T^2\right\}$, it can be seen that even with the modification of $L'$ as in our case, $L + L'$ remains of the same order. Also, the term $\log LL'$ only undergoes a constant scaling. Hence, the change in $L'$ in this case does not affect the rate of estimation error for dilated CNNs.

Therefore, the rate for $\left(U\left(\mathcal{F}_{2,2}^{\gamma_a}\right)\right)^\infty$ is already known from Theorem 3. Hence, to determine the rate for $U\left(\left(\mathrm{F}_{2,2}\left(\Gamma\right)\right)^\infty\right)$, we only need to consider the $\max_{a \in \Gamma(Q_\epsilon)}$ of the rate for $\left(U\left(\mathcal{F}_{2,2}^{\gamma_a}\right)\right)^\infty$.

From Theorem 3, we have:

$$\mathbb{E}_{P^n}\left[\left\|\hat{f} - f^\circ\right\|_{P_X}^2\right] \lesssim \max_{a \in \Gamma(Q_\epsilon)} n^{-\frac{(2-r)a_{i_1}}{2a_{i_1}+1}} (\log n)^{\frac{2}{\eta}+6}$$

$$= n^{-\frac{(2-r)\underline{a}}{2\underline{a}+1}} (\log n)^{\frac{2}{\eta}+6},$$

where, in the last step, we have used the fact that $a_{i_j} \geq \underline{a}j^\eta$ based on the definition of $\Gamma$.

The same reasoning can be applied when considering $\Gamma = \Gamma\left(Q_\epsilon^a\right)$, $\gamma_a\left(s\right) = \max_i\{a_i s_i\}$, and $a^\star = \tilde{a}$. ∎

## D.2 Proof of Theorem 7

*Proof of Theorem 7.* Let us consider the case where we define $\Gamma = \Gamma\left(Q_\epsilon^m\right)$, $\gamma_a\left(s\right) = \langle a, s \rangle$, and $a^\star = \underline{a}$. The proof for the other case follows a similar argument and is omitted.

First, based on the assumptions of this theorem, Theorem 4 and Corollary 6 hold. This means that the convergence rate of the estimation error using the linear estimator is $n^{-\frac{2\underline{a}}{2\underline{a}+1+c}}$, while for the dilated CNNs, it is $n^{-\frac{(2-r)\underline{a}}{2\underline{a}+1}}$. However, we only consider the polynomial order part of $n$ for comparison with the linear estimator, as the polylogarithmic order terms can be neglected when $n$ is sufficiently large.

Now, for the extended CNN to dominate the linear estimator, the following condition must hold:

$$n^{-\frac{(2-r)\underline{a}}{2\underline{a}+1}} < n^{-\frac{2\underline{a}}{2\underline{a}+1+c}}.$$

By elementary algebraic manipulations, we can derive $c > \frac{(2\underline{a}+1)r}{2-r}$. However, note that the assumption $0 < c < 2\underline{a} - 1$ imposes a constraint on $c$. From this, we can infer that in order for $c$ to exist, we need $\frac{(2\underline{a}+1)r}{2-r} < 2\underline{a} - 1 \iff r < \frac{2\underline{a}-1}{2\underline{a}}$ to hold. ∎

