{2\left(a^\dagger - v\right)}{2\left(a^\dagger - v\right)+1}} \left(\log n\right)^{\frac{2}{\eta}+2} \max\left\{ \left(\log n\right)^{\frac{4}{\eta}}, \left(\log n\right)^4 \right\}$$

$$\lesssim d_{\text{out}} n^{-\frac{2\left(a^\dagger - v\right)}{2\left(a^\dagger - v\right)+1}} \left(\log n\right)^{\frac{2}{\eta}+2} \max\left\{ \left(\log n\right)^{\frac{4}{\eta}}, \left(\log n\right)^4 \right\}.$$

Here, Okumoto & Suzuki (2021) choose $T$ such that $T = \frac{a^\dagger}{2(a^\dagger - v)+1} \log_2 n$. Furthermore, from Theorem 1, we have $d_{\text{out}} = (B_2)^2 2^{r\left(1-\frac{v}{a^\dagger}\right)T}$. Substituting these values into the above equation, we obtain:

$$\mathbb{E}_{P^n}\left[ \left\| \hat{f} - f^\circ \right\|_{P_X}^2 \right] \lesssim n^{-\frac{(2-r)\left(a^\dagger - v\right)}{2\left(a^\dagger - v\right)+1}} \left(\log n\right)^{\frac{2}{\eta}+2} \max\left\{ \left(\log n\right)^{\frac{4}{\eta}}, \left(\log n\right)^4 \