# OpenReview forum: "Minimax optimality of convolutional neural networks for infinite dimensional input-output problems and separation from kernel methods"
_ICLR.cc/2024/Conference — ICLR 2024 poster_

### Official Review · Reviewer_DeXd · 2023-11-01

**Soundness:** 3 good
**Presentation:** 3 good
**Contribution:** 3 good
**Rating:** 8
**Confidence:** 2

**Summary:**

In this paper, the authors study the setting of estimation in infinite dimensional input and output using dilated CNNs. They derive approximation and estimation errors when the true function satisfies certain smoothness type conditions. Additionally, they show that these CNNs achieve the minimax optimal rate for estimation accuracy. Finally, they also show that dilated CNNs outperform other models for infinite dimensional estimation like kernel ridge regression and k-NN again in a minimax sense.

**Strengths:**

1. The main theoretical results appear to be novel, and are a significant improvement over previous results.
2. The convergence rate of the estimation error of the dilated CNNs is also shown to be minimax optimal up to poly-log factors under a specific regime.
3. Further, the paper also contains theoretical results which show that dilated CNNs outperform linear estimators, which seems to provide some theoretical justification for the successes of deep learning in high-dimensional spaces.
4. The paper is well written and the proofs seem correct.

**Weaknesses:**

1. The results are specific to the dilated CNN model, and it is not clear how to extend them other model classes.

**Questions:**

1. Can the authors provide some more intuition and insight on how the parameter $r$ affects the class of functions being studied?
2. As a followup, what is the motivation behind where assumption 3 comes from, and can you say anything about example problems in practice that satisfy this assumption?

---

> ### Author Response · Authors · 2023-11-19
>
> Thank you for your detailed review amidst your busy schedule.
>
> **Q:** The results are specific to the dilated CNN model, and it is not clear how to extend them other model classes.
>
> **A:**
> As you correctly pointed out, in this discussion, we are capitalizing on the unique structure of dilated CNNs. This is not limited to dilated CNNs alone. For instance, in Takakura & Suzuki (2023), there was a discussion on the estimation error for Transformers, where they focused on the unique structure of Transformers, such as self-attention, for their proof. While it is true that the discussion may be somewhat limited, conversely, it can be said that a theoretical proof utilizing the unique computations of an architecture provides deep insights and considerations into that architecture. In other words, such discussions are attempts to grasp the essence of the architecture and are, therefore, thought to be beneficial.
>
> **Q:** Can the authors provide some more intuition and insight on how the parameter $r$ affects the class of functions being studied?
>
> **A:**
> In this context, $r$ is a parameter defining the rate at which each output converges to $0$ as its dimension increases. Please see Assumption 3 (3) for reference. As $r$ approaches $0$, the penalty becomes stronger, and the output setting becomes almost identical to that of a one-dimensional scenario. This is the setting described in Okumoto & Suzuki (2021), and indeed, their rate also converges in the limit as $r \to 0$. Thus, $r$ is simply a parameter that determines the convergence speed of the output of the function being studied.
>
> **Q:** As a followup, what is the motivation behind where assumption 3 comes from, and can you say anything about example problems in practice that satisfy this assumption?
>
> **A:**
> Let's consider the audio-to-audio setting, as mentioned in the paper. Also, since we are dealing with a $\gamma$-smooth space, it is assumed that the data is decomposed into the frequency domain. Generally, for typical audio data, the energy decreases as the frequency band becomes higher. Moreover, since the maximum of human audible frequency is considered to be 20,000 Hz, information beyond that is unnecessary. From these facts, setting $r$ as in this study to gradually attenuate the output is consistent with real-world practical applications, in my opinion.

---

### Official Review · Reviewer_Ex9R · 2023-11-01

**Soundness:** 3 good
**Presentation:** 3 good
**Contribution:** 3 good
**Rating:** 6
**Confidence:** 2

**Summary:**

The paper explores the optimality of dilated CNNs in estimating mappings between infinite-dimensional input and output spaces. Through analysis of their approximation and estimation capabilities, the authors establish that the accuracy of approximation and estimation errors is influenced by the smoothness and decay rate of the output relative to the infinity norm. They also provide evidence showing that dilated CNNs outperform linear estimators like kernel ridge regression and k-NN estimators in terms of minimizing estimation errors. This finding emphasizes the efficacy of feature learning accomplished by deep neural networks.

**Strengths:**

1. The paper provides a thorough analysis of the approximation and estimation abilities of dilated CNNs for estimating mappings between infinite-dimensional input and output spaces.

2. The authors demonstrate that the estimation accuracy achieved by dilated CNNs aligns with the minimax optimal rate of convergence.

3. The authors show that dilated CNNs are adaptive to the unknown smoothness structure.

**Weaknesses:**

1. The absence of empirical validation on real-world datasets weakens the practical relevance of the findings

2. The paper briefly mentions potential future directions for research but does not thoroughly discuss the limitations of the study

**Questions:**

1. Assumption 3 seems to be a little bit strong. Could you provide a specific example to illustrate it more concretely?

2. The computational aspect can pose challenges due to the infinite-dimensional nature of both the input and output, particularly when dealing with large-scale data.

3. Can the dilated CNN be used to learn linear operator? Will the estimation rate be improved?

---

> ### Author Response · Authors · 2023-11-19
>
> Thank you for your detailed review amidst your busy schedule.
>
> **Q:** The absence of empirical validation ...
>
> **A:**
> Thank you for your comment. First, in the current setting, we are dealing with a $\gamma$-smooth space, which includes infinite-dimensional input-output functions that possess distinct smoothness along each axis. This means that it links well with practical problems involving data like images or audios, which can be decomposed into a frequency domain.
> We believe that these analyses facilitate our understanding on the success of deep learning.
> However, this remains a generalized discussion, and its applicability to specific cases is a subject for future research. Indeed, conducting simple numerical experiments to verify its effectiveness is possible. We will consider this for future work.
>
> **Q:** The paper briefly mentions ...
>
> **A:**
> Indeed, as you pointed out, our paper only briefly touches upon the limitations of our research. One such limitation is the dependence on $p \geq 2$. For example, the minimax optimal rate discussed in Theorem 2 also incorporates this assumption. This arises directly from Lemma 11, specifically from the evaluation of $\delta_s (f)$. The difficulty in evaluating $\delta_s (f)$ has also impacted prior research. For instance, in Okumoto & Suzuki (2022), which presented the approximation error of one-dimensional output in dilated CNNs, the term $v$ used in our study is incorporated due to this evaluation. In this study, we consistently used the assumption $p \geq 2$ to align with previous research, but revisiting this assumption will be part of our future work.
> According to your comment, we have added a section on the limitation of our paper, which we believe makes our paper more scientifically fair.
>
> **Q:** Assumption 3 seems to ...
>
> **A:**
> Assumption 3 includes two sub-assumptions, (2) and (3). Regarding (2), it is indeed possible to weaken it further. In this paper, for simplicity, we assumed that each output's target function falls within a $\gamma$-smooth space with the same parameters $p, q$, which is a somewhat limiting situation. For instance, relaxing this assumption to allow multiple parameters $p, q$ is feasible. This would increase the number of bases required for the FNNs needed for representation. Investigating how this affects the rate is a task for future research. As for (3), weakening the L2-norm further is mathematically challenging. However, these assumptions seem natural when considering practical applications, so we believe they do not pose a problem.
>
> **Q:** The computational aspect can pose ...
>
> **A:**
> In this research, we have not specifically discussed computational complexity, making it difficult to address that aspect.
> However, our theoretical analysis gives a theoretical guarantee of the approximation error induced by truncating the input and output at a finite length.
> This enables us to implement the real deep neural networks entirely on the finite dimensional space with the approximation error guarantee, which also addresses the computational issue.
> We believe that a part of the success of deep learning is due to the fact that they approximate the input-output space by a sufficiently high dimensional space.
> Interestingly, we have achieved a rate independent of the length of input and output, which are infinite-dimensional. This means that the study provides a theoretical guarantee for the convergence of learning on long input and output data that can be considered as infinite-dimensional.
>
> **Q:** Can the dilated CNN be used to ...
>
> **A:**
> In this paper, we discussed operators within the $\gamma$-smooth space.
> Theoretically, the class of linear operators is included in the extreme case $\gamma(s)  \to \infty$. Hence, we may naively apply our theoretical analysis to such a situation by considering arbitrary large $(a_i)_{i+1}^\infty$ and will achieve at least sub-optimal rate.
> However, it is not trivial whether we can obtain an optimal rate by naively taking the limit of $a_i \to \infty$ because the constants depending these parameters can blow up.
> Hence, we consider the linear operator estimation problem as a different line of research.
> The estimation error of linear operators, for instance, is discussed in Jin (2022). In this work, the input and output spaces are considered as reproducing kernel Hilbert spaces (RKHS), and the linear operator between them is targeted for approximating and estimating errors. However, this study utilizes the spectral decomposition of the covariance operator to make RKHS more manageable and develops a discussion on the upper bound based on that. Thus, to discuss our CNNs in this setting, a different line of argument is required.

---

### Official Review · Reviewer_qUYW · 2023-11-03

**Soundness:** 4 excellent
**Presentation:** 4 excellent
**Contribution:** 3 good
**Rating:** 8
**Confidence:** 3

**Summary:**

In this paper, the authors consider the regression problem with infinite dimensional input and output. This setting  is motivated by recent applications such as image to text mapping.  For their target function, they assume that it belongs to certain gamma smooth spaces, which can be thought as extensions of mixed Besov spaces and anisotropic Besov spaces to infinite dimension output.  They consider learning this class of functions using dilated CNNs, which consists of multilayer convolutions followed by fully connected neural networks. Assuming that the ERM can be constructed, they characterize the approximation and estimation rates, and show that dilated CNNs achieve the minimax optimal rate for norm $p \geq 2$. They further prove a lower bound for linear estimator, which shows that dilated CNNs have better rates under some conditions.

**Strengths:**

- This paper belongs to a long tradition that has sought to study neural networks by decoupling the statistical from the computational aspects. By considering directly the properties of the empirical risk minimizers, one can probe the adaptivity property of neural networks by studying their decay rates for different function classes.  Within this context, the authors do a serious job in carefully constructing the function target class and deriving tight statistical rates.
- The paper is well written and clear despite the amount of technical notations.

**Weaknesses:**

- As a general criticism of this line of work (not particular to this paper), it is unclear how much it informs practical neural networks. However, one can consider these results to be an important background for the study of neural networks.

**Questions:**

- Is there no activation in the convolutional layer?
- What does non-adaptive approach/adaptive approach refer to in section 4.1 and in the conclusion? I guess, this would improve the dependency over the width.
- As a personal taste, I would replace the last line of the abstract "explains the success of deep learning", by a sentence of the type "provide a theoretical basis for understanding the success".

---

> ### Author Response · Authors · 2023-11-19
>
> Thank you for your detailed review amidst your busy schedule.
>
> **Q:** As a general criticism of this line of work (not particular to this paper), it is unclear how much it informs practical neural networks. However, one can consider these results to be an important background for the study of neural networks.
>
> **A:**
> As you pointed out, the extent to which this can be applied to practical problems varies depending on the framework being handled. In this setting, we are dealing with a $\gamma$-smooth space, which includes infinite-dimensional input-output functions that have different smoothness along each axis. This means that it is well linked to practical problems that handle data like images or audios, which can be decomposed into a frequency domain. However, this remains a generalized discussion, and whether it can be applied to specific cases is a subject for future research.
> Indeed, we may consider a different characterization of ``importance'' of each coordinate by changing the scale of each coordinate like $x_i \in [0,r_i]$ for $(r_i)_{i=1}^\infty$ representing the rage of each coordinate. We would like to defer it to the future work.
>
> **Q:** Is there no activation in the convolutional layer?
>
> **A:**
> Yes, there is not nonlinear activation in the convolutional layers.
> However, we would like to point out that the linear activation can be realized by neural network with the nonlinear ReLU activation.
> The convolution layer in this instance simply performs a convolution using a certain filter. For example, in the proof, in the simplest case (where $\gamma$ has mixed smoothness and $a$ is a monotonically increasing sequence of positive real numbers), the convolution layer serves the role of extracting a part of the infinite-dimensional input and passing it on to the subsequent FNNs.
>
> **Q:** What does non-adaptive approach/adaptive approach refer to in section 4.1 and in the conclusion? I guess, this would improve the dependency over the width.
>
> **A:**
> The term ``adaptive'' used here is commonly employed in the theory of deep learning as well as the theoretical statistics, meaning that it can perform feature selection adaptively in response to data. In the case of this example, dilated CNNs can learn adaptively from the data even when $a$ has sparsity. For instance, when $\gamma$ has mixed smoothness, they can achieve an error rate that depends solely on the minimum of $a_i$, corresponding to the most important features. For a detailed theoretical proof of this, please refer to Okumoto and Suzuki (2021).
>
> **Q:** As a personal taste, I would replace the last line of the abstract "explains the success of deep learning", by a sentence of the type "provide a theoretical basis for understanding the success".
>
> **A:**
> Thank you for your comment. As discussed in your initial point, it is indeed true that there exists a gap between theory and practical application. Therefore, softening the phrasing as suggested would be a very good improvement.
> Following your suggestion, we fixed our abstract.

---

### Meta-Review · Area_Chair_o8Dm · 2023-12-07

**Metareview:**

The paper explores the optimality of dilated CNNs in estimating mappings between infinite-dimensional input and output spaces. Through analysis of their approximation and estimation capabilities, the authors establish that the accuracy of approximation and estimation errors is influenced by the smoothness and decay rate of the output relative to the infinity norm. They also provide evidence showing that dilated CNNs outperform linear estimators like kernel ridge regression and k-NN estimators in terms of minimizing estimation errors. This finding emphasizes the efficacy of feature learning accomplished by deep neural networks. All reviewers agree that this paper makes important contributions to understanding the statistical properties of CNN. The AC also agrees and thus recommends acceptance.

**Justification For Why Not Higher Score:**

This paper only studies the statistical properties. It is still unclear whether the gradient-based algorithm can solve the empirical risk problem.

**Justification For Why Not Lower Score:**

The statistical analyses are quite comprehensive.

---

### Decision · Program_Chairs · 2024-01-16

Accept (poster)